# Accurate medium-range global weather forecasting with 3D neural networks

Kaifeng Bi[1], Lingxi Xie[1], Hengheng Zhang[1], Xin Chen[1], Xiaotao Gu[1] & Qi Tian[1✉]

Weather forecasting is important for science and society. At present, the most accurate forecast system is the numerical weather prediction (NWP) method, which represents atmospheric states as discretized grids and numerically solves partial differential equations that describe the transition between those states[1]. However, this procedure is computationally expensive. Recently, artificial-intelligence-based methods[2] have shown potential in accelerating weather forecasting by orders of magnitude, but the forecast accuracy is still significantly lower than that of NWP methods. Here we introduce an artificial-intelligence-based method for accurate, medium-range global weather forecasting. We show that three-dimensional deep networks equipped with Earth-specific priors are effective at dealing with complex patterns in weather data, and that a hierarchical temporal aggregation strategy reduces accumulation errors in medium-range forecasting. Trained on 39 years of global data, our program, Pangu-Weather, obtains stronger deterministic forecast results on reanalysis data in all tested variables when compared with the world's best NWP system, the operational integrated forecasting system of the European Centre for Medium-Range Weather Forecasts (ECMWF)[3]. Our method also works well with extreme weather forecasts and ensemble forecasts. When initialized with reanalysis data, the accuracy of tracking tropical cyclones is also higher than that of ECMWF-HRES.

Weather forecasting is an important application of scientific computing that aims to predict future weather changes, especially in regards to extreme weather events. In the past decade, high-performance computing systems have greatly accelerated research in the field of numerical weather prediction (NWP) methods[1]. Conventional NWP methods are primarily concerned with describing the transitions between discretized grids of atmospheric states using partial differential equations (PDEs) and then solving them with numerical simulations[4–6]. These methods are often slow; a single simulation for a ten-day forecast can take hours of computation in a supercomputer that has hundreds of nodes[7]. In addition, conventional NWP algorithms largely rely on parameterization, which uses approximate functions to capture unresolved processes, where errors can be introduced by approximation[8,9].

The rapid development of deep learning[10] has introduced a promising direction, which the scientific community refers to as artificial intelligence (AI)-based methods[2,11–16]. Here, the methodology is to train a deep neural network to capture the relationship between the input (reanalysis weather data at a given point in time) and the output (reanalysis weather data at the target point in time). On specialized computational devices such as graphics processing units (GPUs), AI-based methods are extremely fast. To give a recent example, FourCastNet[2] takes only 7 s to compute a 100-member, 24-hour forecast, which is orders of magnitudes faster than conventional NWP methods. However, the accuracy of FourCastNet is still below satisfactory; its root mean square error (RMSE) of a 5-day Z500 (500 hPa geopotential) forecast is 484.5, which is much worse than the 333.7 reported by the operational integrated forecasting system (IFS) of the European Centre

for Medium-Range Weather Forecasts (ECMWF)[3]. In a recent survey[17], researchers agreed that AI holds great potential, but admitted that "a number of fundamental breakthroughs are needed" before AI-based methods can beat NWP.

These breakthroughs seem to be happening earlier than expected. Here we present Pangu-Weather (see Methods for an explanation of the name 'Pangu'), a powerful AI-based weather forecasting system that produces stronger deterministic forecast results than the operational IFS on all tested weather variables against reanalysis data. Our technical contributions are two-fold. First, we integrated height information into a new dimension so that the input and output of our deep neural networks can be conceptualized in three dimensions. We further designed a three-dimensional (3D) Earth-specific transformer (3DEST) architecture to inject Earth-specific priors into the deep networks. Our experiments show that 3D models, by formulating height into an individual dimension, have the ability to capture the relationship between atmospheric states in different pressure levels and thus yield significant accuracy gains, compared with two-dimensional models such as FourCastNet[2]. Second, we applied a hierarchical temporal aggregation algorithm that involves training a series of models with increasing forecast lead times. Hence, in the testing stage, the number of iterations needed for medium-range weather forecasting was largely reduced, and the cumulative forecast errors were alleviated. Experiments on the fifth generation of ECMWF reanalysis (ERA5) data[18] validated that Pangu-Weather is good at deterministic forecast and extreme weather forecast while being more than 10,000-times faster than the operational IFS.

[1]Huawei Cloud, Shenzhen, China. ✉e-mail: tian.qi1@huawei.com

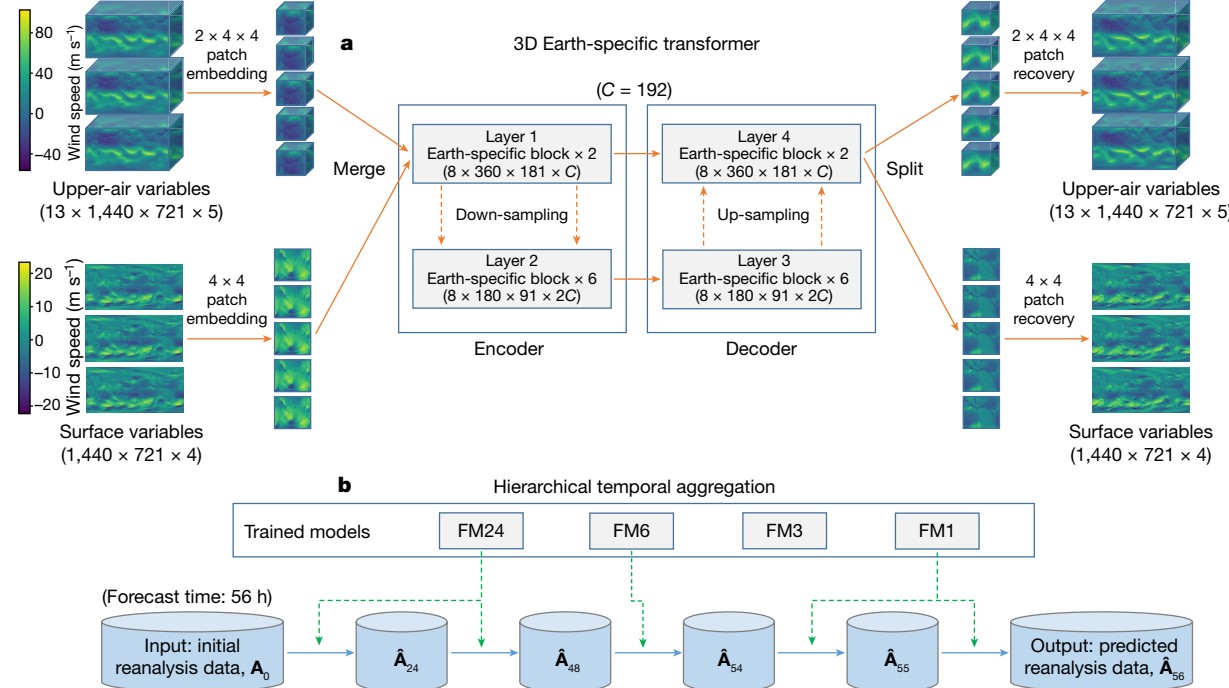

**Fig. 1 | Network training and inference strategies. a**, 3DEST architecture. Based on the standard encoder–decoder design of vision transformers, we adjusted the shifted-window mechanism[19] and applied an Earth-specific positional bias. **b**, Hierarchical temporal aggregation. Once given a lead time,

we used a greedy algorithm to perform forecasting with as few steps as possible. We use FM1, FM3, FM6 and FM24 to indicate the forecast models with lead times being 1 h, 3 h, 6 h and 24 h, respectively. $\mathbf{A}_0$ is the input weather state and $\hat{\mathbf{A}}_t$ denotes the predicted weather state at time $t$ (in hours).

## Global weather forecasting with 3D networks

We established our weather forecast system via deep learning. The methodology involves training deep neural networks to take reanalysis weather data at a given point in time as input, and then produce reanalysis weather data at a future point in time as output. We used a single point in time for both input and output. The time resolution of the ERA5 data is 1 h; in the training subset (1979–2017), there were as many as 341,880 time points, the amount of training data in one epoch. To alleviate the risk of over-fitting, we randomly permuted the order of sample from the training data at the start of each epoch. We trained four deep networks with lead times (the time difference between input and output) at 1 h, 3 h, 6 h and 24 h, respectively. Each of the four deep networks was trained for 100 epochs, and each of them takes approximately 16 days on a cluster of 192 NVIDIA Tesla-V100 GPUs.

The architecture of our deep network is shown in Fig. 1a. This architecture is known as the 3D Earth-specific transformer (3DEST). We fed all included weather variables, including 13 layers of upper-air variables and the surface variables, into a single deep network. We then performed patch embedding to reduce the spatial resolution and combined the down-sampled data into a 3D cube. The 3D data are propagated through an encoder–decoder architecture derived from the Swin transformer[19], a variant of a vision transformer[20], which has 16 blocks. The output is split into upper-air variables and surface variables and is up-sampled with patch recovery to restore the original resolution. To inject Earth-specific priors into the deep network, we designed an Earth-specific positional bias (a mechanism of encoding the position of each unit; detailed in Methods) to replace the original relative positional bias of Swin. This modification increases the number of bias parameters by a factor of 527, with each 3D deep network containing approximately 64 million parameters. Compared with the baseline, however, 3DEST has the same computational cost and has a faster convergence speed.

The lead time of a medium-range weather forecast is 7 days or longer. This prompted us to call the base deep networks (lead times being 1 h, 3 h, 6 h or 24 h) iteratively, using each forecasted result as the input of the next step. To reduce the cumulative forecast errors, we introduced hierarchical temporal aggregation, a greedy algorithm that always calls for the deep network with the largest affordable lead time. Mathematically, this greatly reduces the number of iterations. As an example, when the lead time was 56 h, we would execute the 24-hour forecast model 2 times, the 6-hour forecast model 1 time, and the 1-hour forecast model 2 times (Fig. 1b). Compared with FourCastNet[2], which uses a fixed 6-hour forecast model, our method is faster and more accurate. The limitation of this strategy is discussed in Methods.

## Experimental setting and main results

We evaluated Pangu-Weather on the ERA5 data[18], which is considered the best known estimation for most atmospheric variables[21,22]. To fairly compare Pangu-Weather against FourCastNet[2], we trained our 3D deep networks on 39 years of data (from 1979 to 2017), validated them on 2019 data and tested them on 2018 data. We studied 69 factors, including 5 upper-air variables at 13 pressure levels (50 hPa, 100 hPa, 150 hPa, 200 hPa, 250 hPa, 300 hPa, 400 hPa, 500 hPa, 600 hPa, 700 hPa, 850 hPa, 925 hPa and 1,000 hPa) and 4 surface variables. When tested against reanalysis data, for each tested variable, Pangu-Weather produces a lower RMSE and a higher anomaly correlation coefficient (ACC) than the operational IFS and FourCastNet, the best NWP and AI-based methods, respectively. In particular, with a single-member forecast, Pangu-Weather reports an RMSE of 296.7 for a 5-day Z500 forecast, which is lower than that for the operational IFS and FourCastNet, which reported 333.7 and 462.5, respectively. In addition, the inference cost of Pangu-Weather is 1.4 s on a single GPU, which is more than 10,000-times faster than the operational IFS and on par with FourCastNet. Pangu-Weather not only produces strong quantitative results (for example, RMSE and ACC) but also preserves

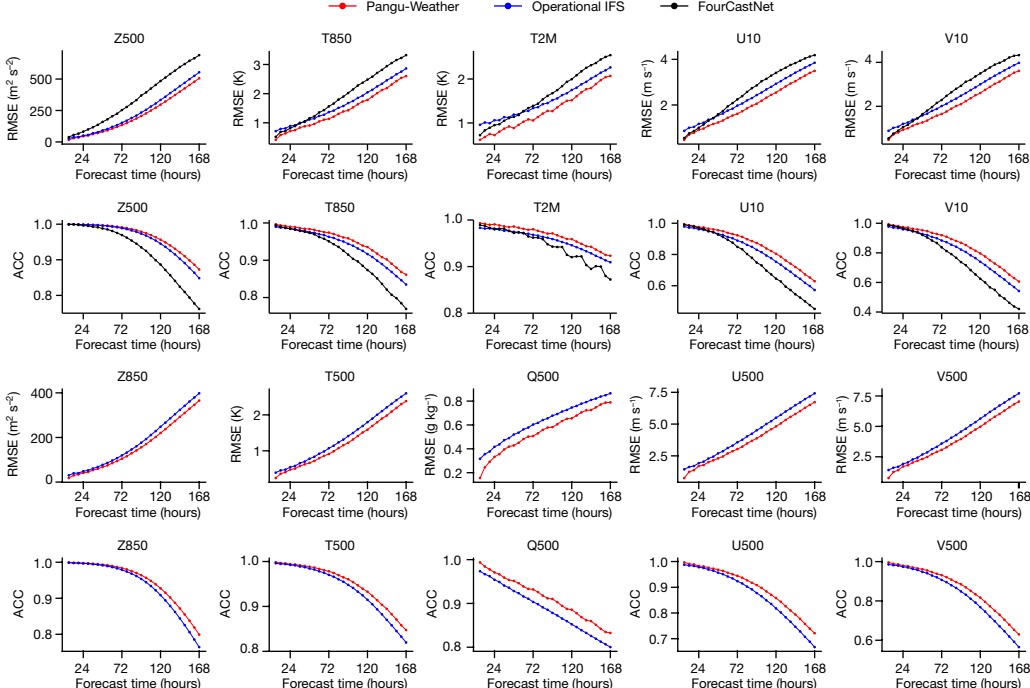

**Fig. 2 | Pangu-Weather produces higher accuracy than the operational IFS and FourCastNet in deterministic forecasts on the ERA5 data.** Ten variables were compared in terms of latitude-weighted RMSE (lower is better) and ACC (higher is better), where the first five variables were reported in FourCastNet and the last five were not. Here, Z500, T500, Q500, U500 and V500 indicate the geopotential, temperature, specific humidity, and the $u$-component and $v$-component of wind speed at 500 hPa, respectively. Z850 and T850 indicate the geopotential and temperature at 850 hPa, respectively. T2M indicates the 2-m temperature, and U10 and V10 indicate the $u$-component and $v$-component of 10-m wind speed, respectively.

sufficient details for investigating certain extreme weather events. To demonstrate this capability, we studied the important application of tropical cyclone tracking. By finding the local minimum of mean sea-level pressure (MSLP), one of the surface variables, our algorithm achieved high accuracy in tracking 88 named tropical cyclones in 2018, including some (for example, Typhoon Kong-rey and Typhoon Yutu) that remain a challenge for the world's best tracking systems, such as ECMWF-HRES (where HRES stands for high-resolution). Our research sheds light on AI-based medium-range weather forecasting systems and advances the progress on the path towards establishing AI as a complement to or surrogate for NWP, an achievement that was previously thought to be far off in the future[17].

## Deterministic global weather forecast

We performed the deterministic forecasting on the unperturbed initial states from ERA5. We then compared Pangu-Weather to the strongest methods in both NWP and AI, namely the operational IFS of ECMWF (data downloaded from the TIGGE (THORPEX Interactive Grand Global Ensemble) archive[3]) and FourCastNet[2]. The spatial resolution of Pangu-Weather, 0.25° × 0.25°, was determined by the training data, which is comparable to the control forecast of ECMWF ENS[5] and identical to FourCastNet. The spacing of the forecast (the minimal unit of forecast time) of Pangu-Weather is 1 h, 6 times less than FourCastNet.

The overall forecast results for 2018 are shown in Fig. 2. For each tested variable, including upper-air and surface variables, Pangu-Weather reports more accurate results than both the operational IFS and Four-CastNet (when the variable is reported). In terms of RMSE (lower is better), Pangu-Weather typically reports 10%-lower values than operational IFS and 30%-lower values than FourCastNet. The advantage persists across all lead times (from 1 h to 168 h, that is, 7 days), and for some variables such as Z500, the advantage becomes more significant with a greater lead time. For quantitative studies in the Northern Hemisphere,

the Southern Hemisphere and the tropics, refer to the Extended Data Figs. 1–3. For the forecast results for 2020 and 2021 and the comparison with the results for 2018, refer to Extended Data Fig. 4.

To demonstrate our advantage, we introduced a concept called 'forecast time gain', which corresponds to the average difference between the lead times of Pangu-Weather and a competitor when they report the same accuracy. Pangu-Weather typically shows a forecast time gain of 10–15 h over the operational IFS, and for some variables such as specific humidity, the gain is more than 24 h. This implies the difficulty that conventional NWP methods have when forecasting specific variables, yet AI-based methods benefit by learning effective patterns from an abundance of training data. Compared with FourCastNet, the forecast time gain of Pangu-Weather is as great as 40 h, showing the significant advantage of our technical design, resulting especially from the 3D deep networks and the advanced temporal aggregation strategy. The forecast time gains of Pangu-Weather in terms of different weather variables are summarized in Extended Data Table 2.

Figure 3 shows a visualization of the 3-day forecast results of Pangu-Weather. We studied two upper-air variables, Z500 and T850 (850 hPa temperature), and two surface variables, 2-m temperature and 10-m wind speed, and compared the results with the operational IFS and the ERA5 ground truth. The results of both Pangu-Weather and operational IFS are sufficiently close to the ground truth, yet there are visible differences between them. Pangu-Weather produced smoother contour lines, implying that the model tends to forecast similar values for neighbouring regions. It is a general property of any regression algorithm (including deep neural networks) to converge on average values. In contrast, the operational IFS forecast is less smooth, because it calculates a single estimated value at each grid cell by solving a system of PDEs with initial conditions, while the chaotic nature of weather and the inevitably imprecise knowledge of the initial conditions and sub-grid scale processes can cause statistical uncertainties in each forecast.

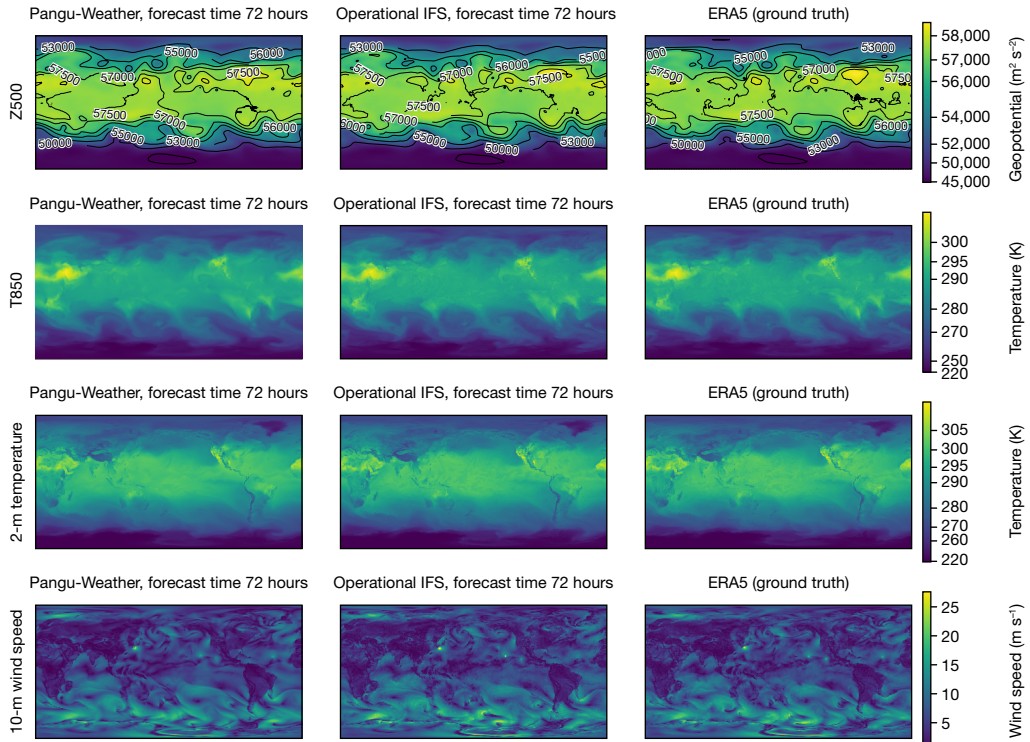

**Fig. 3 | Visualization of forecast results.** The 3-day forecast of two upper-air variables (Z500 and T850) and two surface variables (2-m temperature and 10-m wind speed). For each case, Pangu-Weather (left), the operational IFS[3] (middle) and the ERA5 ground truth[18] (right) are shown. For all cases, the input time is 00:00 UTC on 1 September 2018.

## Tracking tropical cyclones

Next, we used Pangu-Weather to track tropical cyclones. Given an initial time point, we set the lead time to be multiples of 6 h (ref. 23) and initiated Pangu-Weather to forecast future weather states. We looked for the local minimum of MSLP that satisfied certain conditions, such as the cyclone eye. The tracking algorithm is described in the supplementary

material for this paper. We used the International Best Track Archive for Climate Stewardship (IBTrACS) project[24,25], which contains the best available estimations for tropical cyclones.

We compared Pangu-Weather with ECMWF-HRES, a strong cyclone tracking method based on high-resolution (9 km × 9 km) operational weather forecasting. We chose 88 named tropical cyclones in 2018 that appear in both IBTrACS and ECMWF-HRES. As shown in Fig. 4,

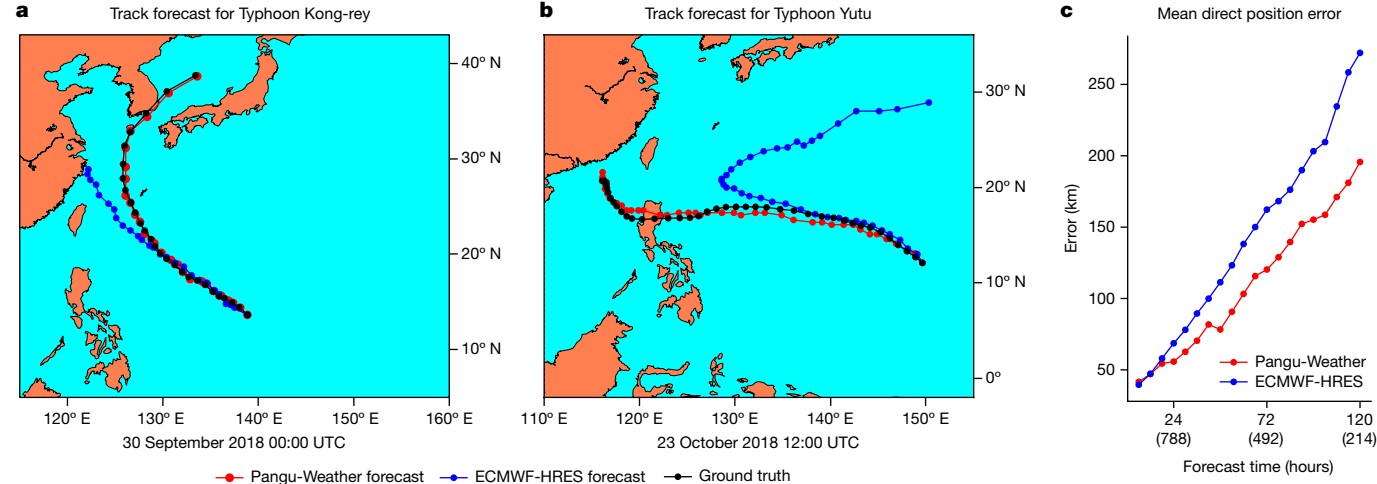

**Fig. 4 | Pangu-Weather is more accurate at early-stage cyclone tracking than ECMWF-HRES. a,b**, Tracking results for two strong tropical cyclones in 2018, that is, Typhoon Kong-rey (2018–25) and Yutu (2018–26). The initial time point is shown below each panel. The time gap between neighbouring dots is 6 h. Pangu-Weather forecasts the correct path of Yutu (that is, it goes to the Philippines) at 12:00 UTC on 23 October 2018, whereas ECMWF-HRES obtains the same conclusion 2 days later, before which it predicts that Yutu will make

a big turn to the northeast. **c**, A comparison between Pangu-Weather and ECMWF-HRES in terms of mean direct position error over 88 cyclones in 2018. Each number in brackets in the *x*-axis indicates the number of samples used to calculate the average. For example, '(788)' means that there are in total 788 initial points from which the typhoon lasts for at least 24 hours, and the 788 direct position errors of Pangu-Weather and ECMWF-HRES were averaged into the final results. Panels **a** and **b** were plotted using the Matplotlib Basemap toolkit.

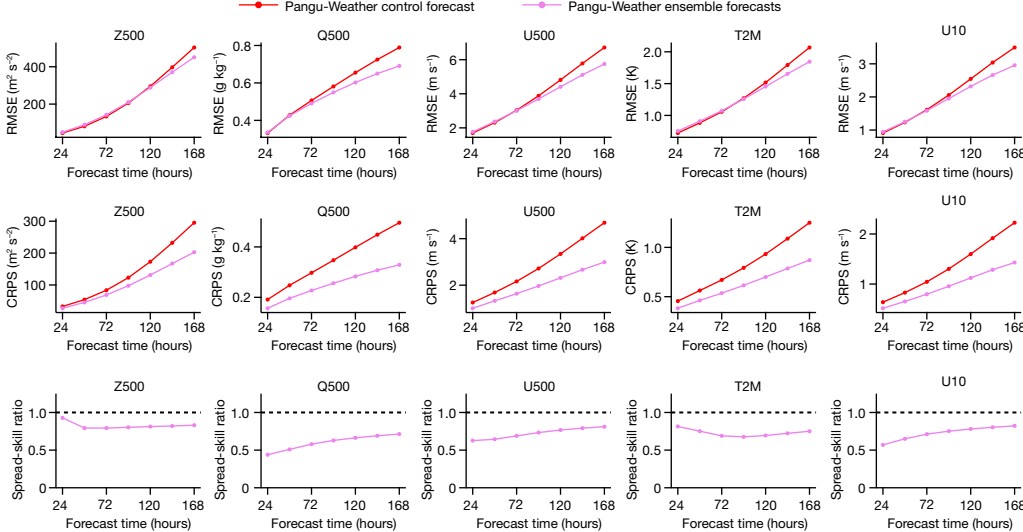

**Fig. 5 | Ensemble forecast results of Pangu-Weather.** The RMSE of the ensemble mean forecast (lower is better) for three upper-air variables (Z500, Q500 and U500) and two surface variables (T2M and U10). We also followed a recent work[35] to plot two metrics, the CRPS (lower is better) and the spread-skill ratio (an ideal ensemble model produces spread-skill ratios of 1.0, shown as the dashed lines), which further demonstrate the properties of our ensemble forecast results. Here, Z500, Q500 and U500 indicate the geopotential, temperature and the *u*-component of wind speed at 500 hPa, respectively. T2M indicates the 2-m temperature and U10 indicates the *u*-component of 10-m wind speed.

Pangu-Weather statistically produced more accurate tracking results than ECMWF-HRES for these cyclones. The 3-day and 5-day mean direct position errors for cyclone eyes were reported at 120.29 km and 195.65 km for Pangu-Weather, which are smaller than 162.28 km and 272.10 km for ECMWF-HRES, respectively. The breakdowns of tracking errors with respect to regions and intensities are provided in Extended Data Fig. 5. The advantage of Pangu-Weather becomes more significant as the lead time increases. We also show the tracking results of the two strongest cyclones in the western Pacific, Kong-rey and Yutu, in Fig. 4. See the supplementary material for a detailed analysis.

Despite the promising tracking results, we point that the direct comparison between Pangu-Weather and ECMWF-HRES is somewhat unfair, because ECMWF-HRES used the IFS initial condition data as its input, whereas Pangu-Weather used reanalysis data.

## Ensemble weather forecast

As an AI-based method, Pangu-Weather is more than 10,000-times faster than the operational IFS. This offers an opportunity for performing large-member ensemble forecasts with small computational costs. We investigated FourCastNet[2] to study a preliminary ensemble method that adds perturbations to initial weather states. We then generated 99 random perturbations (detailed in Methods) and added them to the unperturbed initial state. Thus, we obtained a 100-member ensemble forecast by simply averaging the forecast results. As shown in Fig. 5, for each variable, the ensemble mean is slightly worse than the single-member method in the short-range (for example, 1 day) weather forecasts, but significantly better when the lead time is 5–7 days. This aligns with FourCastNet[2], indicating that large-member ensemble forecasts are especially useful when single-model accuracy is lower, yet they present the risk of introducing unexpected noise to short-range forecasts. Ensemble forecasting presents more benefits to non-smooth variables such as Q500 (500 hPa specific humidity) and U10 (10 m *u*-component of wind speed). In addition, the spread-skill ratio of Pangu-Weather is smaller than 1, indicating that the current ensemble method is somewhat underdispersive. Compared with NWP methods, Pangu-Weather largely reduces the cost of ensemble forecasting, allowing meteorologists to apply their expertise to control noise and improve ensemble forecast accuracy.

## Discussion

In this paper, we present Pangu-Weather, an AI-based system that trains deep networks for fast and accurate numerical weather forecasting. The major technical contributions include the design of the 3DEST architecture and the application of the hierarchical temporal aggregation strategy for medium-range forecasting. By training the models on 39 years of global weather data, Pangu-Weather produces better deterministic forecast results on reanalysis data than the world's best NWP system, the operational IFS of ECMWF, while also being much faster. In addition, Pangu-Weather is excellent at forecasting extreme weather events and performing ensemble weather forecasts. Pangu-Weather reveals the potential of using large pre-trained models for various downstream applications, showing the same trend as other AI scopes, such as computer vision[26,27], natural language processing[28,29], cross-modal understanding[30] and beyond.

Despite the promising forecast accuracy on reanalysis data, our algorithm has some limitations. First, throughout this paper, Pangu-Weather was trained and tested on reanalysis data, but real-world forecast systems work on observational data. There are differences between these data sources; thus, Pangu-Weather's performance across applications needs further investigation. Second, some weather variables, such as precipitation, were not investigated in this paper. Omitting these factors may cause the current model to lack some abilities, for example, making use of precipitation data for the accurate prediction of small-scale extreme weather events, such as tornado outbreaks[31,32]. Third, AI-based methods produce smoother forecast results, increasing the risk of underestimating the magnitudes of extreme weather events. We studied a special case, cyclone tracking, but there is much more work to do. Fourth, temporal inconsistency can be introduced by using models with different lead times. This is a challenging topic worth further investigation.

Looking to the future, there is room for improvement for both AI-based methods and NWP methods. On the AI side, further gains can be found by incorporating more vertical levels and/or atmospheric variables, integrating the time dimension and training four-dimensional deep networks[33,34], using deeper and/or wider networks, or simply increasing the number of training epochs. All of these directions call for more powerful GPU clusters with larger

memories and higher FLOPS (floating point operations per second), which is the current trend of the AI community. On the NWP side, post-processing methods can be developed to alleviate the predictable biases of NWP models. We expect that AI-based and NWP methods will be combined in the future to bring about even stronger performance.

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

# Methods

## Mathematical settings

We denoted all studied global weather variables at time $t$ as $\mathbf{A}_t$. This is a 3D matrix of size $N_{lon} \times N_{lat} \times 69$, where $N_{lon} = 1{,}440$ and $N_{lat} = 721$ are the spatial resolution along the longitude and latitude axes, respectively, and 69 is the number of studied variables. In other words, each horizontal pixel occupies $0.25° \times 0.25°$ on Earth's surface. The mathematical problem is that given the forecast time point $t_0$, assume that $\mathbf{A}_t$ for all $t \le t_0$ are available, the algorithm is asked to predict $\mathbf{A}_{t_0 + \Delta t}$ where $\Delta t$ is called the lead time. Owing to the limitation of GPU memory, in our work, the forecast algorithm only used $\mathbf{A}_{t_0}$ as input and predicted $\mathbf{A}_{t_0 + \Delta t}$ as output. For this purpose, we trained a deep neural network, $f(\mathbf{A}_{t_0}; \theta)$, where $\theta$ denotes the learnable parameters.

**Evaluation metrics.** When the predicted version of $\mathbf{A}_t$ is available ($t = t_0 + \Delta t$), denoted as $\hat{\mathbf{A}}_t$, we computed two metrics, RMSE and ACC, defined as follows:

$$\text{RMSE}(v, t) = \sqrt{\frac{\sum_{i=1}^{N_{lat}} \sum_{j=1}^{N_{lon}} L(i)\, (\hat{\mathbf{A}}_{i,j,t}^{v} - \mathbf{A}_{i,j,t}^{v})^2}{N_{lat} \times N_{lon}}}$$

$$\text{ACC}(v, t) = \sqrt{\frac{\sum_{i=1}^{N_{lat}} \sum_{j=1}^{N_{lon}} L(i)\, \hat{\mathbf{A}}_{i,j,t}^{\prime v} \mathbf{A}_{i,j,t}^{\prime v}}{\sum_{i=1}^{N_{lat}} \sum_{j=1}^{N_{lon}} L(i)\, (\hat{\mathbf{A}}_{i,j,t}^{\prime v})^2 \times \sum_{i=1}^{N_{lat}} \sum_{j=1}^{N_{lon}} L(i)\, (\mathbf{A}_{i,j,t}^{\prime v})^2}}$$

Here, $v$ is any weather variable, $\mathbf{A}_{i,j,t}^{v}$ is a scalar representing the value of $v$ at time $t$ and horizontal coordinate $(i, j)$. $L(i) = N_{lat} \times \frac{\cos \phi_i}{\sum_{i'=1}^{N_{lat}} \cos \phi_{i'}}$ is the weight at latitude $\phi_i$. $\mathbf{A}'$ denotes the difference between $\mathbf{A}$ and the climatology, that is, the long-term mean of weather states that are estimated on the training data over 39 years. The RMSE and ACC values were averaged over all times and horizontal coordinates to produce the average numbers for variable $v$ and lead time $\Delta t$. The RMSE and ACC metrics can also be evaluated for specific regions, for example, in the Northern Hemisphere, the Southern Hemisphere and the tropics. Refer to Fig. 2 and Extended Data Figs. 1–3 for the overall and breakdown results in 2018.

**Ensemble forecast metrics.** We follow a recent work[35] to compute two metrics for ensemble weather forecast, namely, the continuous ranked probability score (CRPS) and the spread-skill ratio (SSR). Mathematically, CRPS is defined as

$$\text{CRPS} = \int_{-\infty}^{+\infty} [F(\hat{\mathbf{A}}_{i,j,t}^{v}) - \mathbb{I}(\mathbf{A}_{i,j,t}^{v} \le z)]\, dz$$

where $F(\cdot)$ denotes the cumulative distribution function of the forecast distribution and $\mathbb{I}(\cdot)$ is an indicator function that takes a value of 1 if the statement is true and 0 otherwise. We follow the original paper and use the xskillscore Python package for computing the CRPS. SSR is obtained by dividing 'spread' by RMSE with spread being

$$\text{Spread}(v, t) = \sqrt{\frac{\sum_{i=1}^{N_{lat}} \sum_{j=1}^{N_{lon}} L(i) \cdot \text{var}(\hat{\mathbf{A}}_{i,j,t}^{v})}{N_{lat} \times N_{lon}}}$$

Here, $\text{var}(\cdot)$ indicates the variance in the ensemble dimension. The spread and RMSE values averaged over all forecasts are used for computing SSR. If an ensemble is perfectly reliable, it should report an SSR of 1.0.

## Data preparation details

The ERA5 dataset[18] contains global, hourly reanalysis data for the past 60 years. The observation data and the prediction of numerical models are blended into reanalysis data using numerical assimilation methods, providing a high-quality benchmark for global weather forecasting. We made use of the reanalysis data of every single hour so that the algorithm can perform hourly weather forecasting. We kept the highest spatial resolution available in ERA5, $0.25° \times 0.25°$ on Earth's sphere, resulting in an input resolution of $1{,}440 \times 721$: the latitude dimension has an extra entry because the northernmost and southernmost positions do not overlap.

We followed WeatherBench[13] to choose 13 out of 37 pressure levels (50 hPa, 100 hPa, 150 hPa, 200 hPa, 250 hPa, 300 hPa, 400 hPa, 500 hPa, 600 hPa, 700 hPa, 850 hPa, 925 hPa and 1,000 hPa) and the surface level. To fairly compare with the online version of the ECMWF control forecast, we chose to forecast the factors published in the TIGGE dataset[3], namely, five upper-air variables (geopotential, specific humidity, temperature, and the $u$-component and $v$-component of wind speed) and four surface variables (2-m temperature, the $u$-component and $v$-component of 10-m wind speed, and MSLP). For a complete list of studied variables and the corresponding abbreviations, refer to Extended Data Table 1. In addition, three constant masks (the topography mask, land–sea mask and soil-type mask) were added to the input of surface variables.

When we prepared for the test data in 2018, we excluded the test points on 1 January 2018 owing to the overlap with training data. In addition, all test points in December 2018 are unavailable for the upper-air variables owing to a server error of ECMWF. FourCastNet also excluded these data from the testing phase.

## Deep network details

There are two sources of input and output data, namely, upper-air variables and surface variables. The former involves 13 pressure levels, each of which has 5 variables, and they together form a $13 \times 1{,}440 \times 721 \times 5$ volume. The latter contains a $1{,}440 \times 721 \times 4$ volume. These parameters were first embedded from the original space into a $C$-dimensional latent space. We used a common technique named patch embedding for dimensionality reduction. For the upper-air part, the patch size is $2 \times 4 \times 4$ so the embedded data have a shape of $7 \times 360 \times 181 \times C$. For the surface variables, the patch size is $4 \times 4$ so the embedded data have a shape of $360 \times 181 \times C$, where $C$ is the base channel width and was set to be 192 in our work. These two data volumes were then concatenated along the first dimension to yield a $8 \times 360 \times 181 \times C$ volume. The volume was then propagated through a standard encoder–decoder architecture with 8 encoder layers and 8 decoder layers. The output of the decoder is still a $8 \times 360 \times 181 \times C$ volume, which was projected back to the original space with patch recovery, producing the desired output. Below, we describe the technical details of each component.

**Patch embedding and patch recovery.** We followed the standard vision transformer to use a linear layer with GELU (Gaussian Error Linear Unit) activation for patch embedding. In our implementation, a patch has $2 \times 4 \times 4$ pixels for upper-air variables and $4 \times 4$ for surface variables. The stride of sliding windows is the same as the patch size, and necessary zero-value padding was added when the data size is indivisible by the patch size. The number of parameters for patch embedding is $(4 \times 4 \times 2 \times 5) \times C$ for upper-air variables and $(4 \times 4 \times 4) \times C$ for surface variables. Patch recovery performs the opposite operation, having the same number of parameters but these parameters are not shared with patch embedding.

**The encoder–decoder architecture.** The data size remains unchanged as $8 \times 360 \times 181 \times C$ for the first 2 encoder layers, whereas for the next 6 layers, the horizontal dimensions were reduced by a factor of 2 and the number of channels was doubled, resulting in a data size of $8 \times 180 \times 91 \times 2C$. The decoder part is symmetric to the encoder part, with the first 6 decoder layers having a size of $8 \times 180 \times 91 \times 2C$ and the next 2 layers having a size of $8 \times 360 \times 181 \times C$. The outputs of the second encoder layer and the seventh decoder layer were concatenated

along the channel dimension. We follow the implementation of Swin transformers[19] to connect the adjacent layers of different resolutions with down-sampling and up-sampling operations. For down-sampling, we merged four tokens into one (the feature dimensionality increases from $C$ to $4C$) and performed a linear layer to reduce the dimensionality to $2C$. For up-sampling, the reverse operations were performed.

**3D Earth-specific transformer.** Each encoder and decoder layer is a 3DEST block. It is similar to the standard vision transformer block[20] but specifically designed to align with Earth's geometry. We used the standard self-attention mechanism of vision transformers. To further reduce computational costs, we inherited the window-attention mechanism[19] to partition the feature maps into windows, each of which contains at most $2 \times 12 \times 6$ tokens. The shifted-window mechanism[19] was applied so that for every layer, the grid partition differs from the previous one by half window size. As coordinates in longitude direction are periodic, the half windows at the left and right edges are merged into one full window. The merge operation was not performed along the latitude direction because it is not periodic. We refer the reader to the original papers[19,20] for more details about vision transformers.

**Earth-specific positional bias.** Swin transformer[19] used a relative positional bias to represent the translation-invariant component of attentions, where the bias was computed upon the relative coordinate of each window. For global weather forecasting, however, the situation is a bit different: each token corresponds to an absolute position on Earth's coordinate system; as the map is a projection of Earth's sphere, the spacing between neighbouring tokens can be different. More importantly, some weather states are closely related to the absolute position. Examples of geopotential, wind speed and temperature are shown in Extended Data Fig. 6. To capture these properties, we introduced an Earth-specific positional bias, which works by adding a positional bias to each token based on its absolute (rather than relative) coordinate.

Mathematically, let the entire feature map be a volume with a spatial resolution of $N_{pl} \times N_{lon} \times N_{lat}$, where $N_{pl}$, $N_{lon}$ and $N_{lat}$ indicate the size along the axes of pressure levels, longitude and latitude, respectively. The data volume was partitioned into $M_{pl} \times M_{lon} \times M_{lat}$ windows, and each window has a size of $W_{pl} \times W_{lon} \times W_{lat}$. The Earth-specific positional bias matrix contains $M_{pl} \times M_{lat}$ submatrices ($M_{lon}$ does not appear here because different longitudes share the same bias: the longitude indices are cyclic and spacing is evenly distributed along this axis), each of which has $W_{pl}^2 \times (2W_{lon} - 1) \times W_{lat}^2$ learnable parameters. When the attention was computed between two units within the same window, we used the indices of the pressure level and latitude, $(m_{pl}, m_{lat})$, to locate the corresponding bias submatrix. Then, we used the intra-window coordinates, $(h_1', \lambda_1', \phi_1')$ and $(h_2', \lambda_2', \phi_2')$, to look up the bias value at $(h_1' + h_2' \times W_{pl}, \lambda_1' - \lambda_2' + W_{lon} - 1, \phi_1' + \phi_2' \times W_{lat})$ of the $(m_{pl}, m_{lat})$th submatrix.

**Design choices.** We briefly discuss other design choices. Owing to the large training overhead, we did not perform exhaustive studies on the hyperparameters and we believe that there exist configurations or hyperparameters that lead to higher accuracy. First, we used $8 (2 + 6)$ encoder and decoder layers, which is significantly fewer than the standard Swin transformer[19]. This is to reduce the complexity of both time and memory. If one has a more powerful cluster with larger GPU memory, increasing the network depth can bring higher accuracy. Second, it is possible to reduce the number of parameters used in the Earth-specific positional bias by parameter sharing or other techniques. However, we did not consider it a key issue, because it is unlikely to deploy the weather forecasting model to edge devices with limited storage. Third, it is possible and promising to feed the weather states of more time indices into the model, which changes all tensors from three dimensions to four dimensions. Although the AI community has shown the effectiveness of four-dimensional deep networks[33,34], the limited available computational budget prevented us from exploring this method.

**Optimization details.** The four individual models were trained for 100 epochs using the Adam optimizer. We used the mean-absolute-error loss. The normalization was performed on each two-dimensional input field (for example, Z500) separately. It worked by subtracting the mean value from the two-dimensional field followed by dividing it by the standard deviation. The mean and standard deviation of each variable were computed on the weather data from 1979 to 2017. The weight for each variable was inversely proportional to the average loss value computed in an early run, which was designed to facilitate equivalence of the contributions by these variables. Specifically, the weights for upper-air variables were 3.00, 0.60, 1.50, 0.77 and 0.54 for Z, Q, T, U and V, respectively, and the weights for surface variables were 1.50, 0.77, 0.66 and 3.00 for MSLP, U10, V10 and T2M, respectively. We added a weight of 1.0 to the mean-absolute-error loss of the upper-air variables and 0.25 to that of the surface variables, and summed up the two losses. We used a batch size of 192 (that is, 1 training sample per GPU). The learning rate started with 0.0005 and gradually annealed to 0 following the cosine schedule. All starting time points in the training subset (1979–2017) were randomly permuted in each epoch to alleviate over-fitting. A weight decay of $3 \times 10^{-6}$ and ScheduledDropPath[36] with a drop ratio of 0.2 were adopted to alleviate over-fitting. We found that all models have not yet arrived at full convergence at the end of 100 epochs, so we expect that extending the training procedure can improve the forecast accuracy. We plotted the accuracy of some tested variables with respect to different lead times (1 h, 3 h, 6 h and 24 h) in Extended Data Fig. 7.

## Inference speed

The inference speed of Pangu-Weather is comparable to that of Four-CastNet[2]. In a system-level comparison, FourCastNet requires 0.28 s for inferring a 24-hour forecast on a Tesla-A100 GPU (312 teraFLOPS), whereas Pangu-Weather needs 1.4 s on a Tesla-V100 GPU (120 tera-FLOPS). Taking GPU performance into consideration, Pangu-Weather is about 50% slower than FourCastNet. Pangu-Weather is more than 10,000-times faster than the operational IFS, which requires several hours in a supercomputer with hundreds of nodes.

## Computation of relative quantile error

We followed a previous work[37] to compare the values of top-level quantiles calculated on the forecast result and ground truth. Mathematically, we set $D = 50$ percentiles, denoted as $q_1, q_2, ..., q_D$. We followed Four-CastNet[2] to set $q_1 = 90\%$ and $q_D = 99.99\%$, and the intermediate percentile values were linearly distributed between $q_1$ and $q_D$ in the logarithmic scale. Then, the corresponding quantiles, denoted as $Q_1, Q_2, ..., Q_D$, were computed individually for each pair of weather variable and lead time. For example, for all 3-day forecasts of the U10 variable, pixelwise values were gathered from all frames for statistics. We followed FourCastNet[2] to plot the extreme percentiles with respect to lead time in Extended Data Fig. 7.

Finally, the relative quantile error (RQE) was computed for measuring the overall difference between the ground truth and any weather forecast algorithm:

$$RQE = \sum_{d=1}^{D} \frac{\hat{Q}_d - Q_d}{Q_d}$$

where $Q_d$ and $\hat{Q}_d$ are the $d$th quantile calculated on the ERA5 ground truth and the forecast algorithm being investigated. RQE can measure the overall tendency, where $RQE < 0$ and $RQE > 0$ imply that the forecast algorithm tends to underestimate and overestimate the intensity of extremes, respectively. We found that both Pangu-Weather and the operational IFS tend to underestimate extremes. Pangu-Weather suffers heavier underestimation as the lead time increases. It is noted that RQE and the individual quantile values have limitations: they do not evaluate whether extreme values occur at the right location and time,

but only look at the value distribution. The ability of Pangu-Weather to capture individual extreme events was further validated with the experiments of tracking tropical cyclones.

## Algorithm for tracking tropical cyclones

We followed a classical algorithm[38] that locates the local minimum of MSLP to track the eye of tropical cyclones. Given the starting time point and the corresponding initial position of a cyclone eye, we iteratively called for the 6-hour forecast algorithm and looked for a local minimum of MSLP that satisfies the following conditions:

- There is a maximum of 850 hPa relative vorticity that is larger than $5 \times 10^{-5}$ within a radius of 278 km for the Northern Hemisphere, or a minimum that is smaller than $-5 \times 10^{-5}$ for the Southern Hemisphere.
- There is a maximum of thickness between 850 hPa and 200 hPa within a radius of 278 km when the cyclone is extratropical.
- The maximum 10-m wind speed is larger than 8 m s$^{-1}$ within a radius of 278 km when the cyclone is on land.

Once the cyclone's eye is located, the tracking algorithm continued to find the next position in a vicinity of 445 km. The tracking algorithm terminated when no local minimum of MSLP is found to satisfy the above conditions. See Extended Data Fig. 8 for two tracking examples.

**Tracking results in different subsets.** We extended Fig. 4c by plotting the mean direct position errors with respect to different basins or different intensities in Extended Data Fig. 5. In each subset, Pangu-Weather reports lower errors and the advantage becomes more significant with a greater lead time, aligning with the conclusions we drew from the entire dataset. Again, we emphasize that the comparison against ECMWF-HRES is somewhat unfair, because ECMWF-HRES used IFS initial condition data, whereas Pangu-Weather used reanalysis data.

**More tropical cyclones.** Below is a more detailed analysis of four tropical cyclones. The advantage of Pangu-Weather mainly lies in tracking cyclone paths in the early stages.
(1) Typhoon Kong-rey (2018-25) is one of the most powerful tropical cyclones worldwide in 2018. As shown in Fig. 4, ECMWF-HRES forecasts that Kong-rey would land in China, but it actually did not. Pangu-Weather, instead, produces accurate tracking results which almost coincide with the ground truth. Also, Extended Data Fig. 8 shows the tracking results of Pangu-Weather and ECMWF-HRES at different time points: the forecast result of Pangu-Weather barely changes with time, and ECMWF-HRES arrives at the conclusion that Kong-rey would not land in China more than 48 h later than Pangu-Weather.
(2) Typhoon Yutu (2018-26) is an extremely powerful tropical cyclone that caused catastrophic destruction in the Mariana Islands and the Philippines. It ties with Kong-rey as the most powerful tropical cyclone worldwide in 2018. As shown in Fig. 4, Pangu-Weather makes the correct forecast result (Yutu goes to the Philippines) as early as 6 days before landing, whereas ECMWF-HRES incorrectly predicts that Yutu will make a big turn to the northeast in the early stage. ECMWF-HRES produces the correct tracking results more than 48 h later than Pangu-Weather.
(3) Hurricane Michael (2018-13) is the strongest hurricane of the 2018 Atlantic hurricane season. As shown in Extended Data Fig. 8, with a starting time that is more than 3 days earlier than landing, both Pangu-Weather and ECMWF-HRES forecast the landfall in Florida. But, the delay of predicted landing time is only 3 h for Pangu-Weather whereas it is 18 h for ECMWF-HRES. In addition, Pangu-Weather shows great advantages in tracking Michael after it landed, whereas the tracking of ECMWF-HRES is shorter and obviously shifts to the east.
(4) Typhoon Ma-on (2022-09) is a severe tropical storm that impacted the Philippines and China. As shown in Extended Data Fig. 8, when the starting time point is about 3 days earlier than the landing,

ECMWF-HRES produces a wrong forecast result that Ma-on would land in Zhuhai, China, whereas the forecast result of Pangu-Weather is close to the truth.

The better tracking results of Pangu-Weather are mainly inherited from the accurate deterministic forecast accuracy on reanalysis data. In Extended Data Fig. 8, we show how Pangu-Weather tracks Hurricane Michael and Typhoon Ma-on following the specified tracking algorithm. Among the four variables, MSLP and 10-m wind speed were directly produced by deterministic forecast, and thickness and vorticity were derived from geopotential and wind speed. This indicates that Pangu-Weather can produce intermediate results that support cyclone tracking, which further assists meteorologists in understanding and exploiting the tracking results.

## Random perturbations

Each perturbation generated for ensemble weather forecast contains 3 octaves of Perlin noise, with the scales being 0.2, 0.1 and 0.05, and the number of periods to generate along each axis (the longitude or the latitude) being 12, 24 and 48, respectively. We used the code provided in a GitHub repository (https://github.com/pvigier/perlin-numpy) and modified the code for acceleration. We added a section to the pseudocode.

## Previous work

There are mainly two lines of research for weather forecasting. Throughout this paper, we have been using 'conventional NWP' or simply 'NWP' methods to refer to the numerical simulation methods, and use 'AI-based' methods to specify data-driven forecasting systems. We understand that, verbally, AI-based methods also belong to NWP, but we followed the convention[17] to use these terminologies.

NWP methods often partition the atmospheric states into discretized grids, use PDEs to describe the transition between them[1,39,40] and solve the PDEs using numerical simulations. The spacing of grids is key to forecast accuracy, but it is constrained by the computational budget and thus the spatial resolution of weather forecasts is often limited. Parameterization[41] is an effective method for capturing unresolved processes. NWP methods have been widely applied, but they are troubled by the super-linearly increasing computational overhead[1,42] and it is often difficult to perform efficient parallelization for them[43]. The heavy computational overhead of NWP also restricts the number of ensemble members, hence weakening the diversity and accuracy of probabilistic weather forecasts.

AI-based methods offer a complementary path for weather forecasting. The cutting-edge technology of AI lies in deep learning[10], which assumes that the complex relationship between input and output data can be learned from abundant training data without knowing the actual physical procedure and/or formulae. In the scope of weather forecasting, AI-based methods were first applied to the problems of precipitation forecasting based on radar data[44–47] or satellite data[48,49], where the traditional methods that are much influenced by the initial conditions were replaced by deep-learning-based methods. The powerful expressive ability of deep neural networks led to success in these problems, which further encouraged researchers to delve into medium-range weather forecasting[2,11–16] as a faster complement or surrogate of NWP methods. State-of-the-art deep-learning methods mostly rely on large models (that is, with large numbers of learnable parameters) to learn complex patterns from the training data.

## The name of 'Pangu'

Pangu is a primordial being and creation figure in Chinese mythology who separated heaven and earth and became geographic features such as mountains and rivers (see https://en.wikipedia.org/wiki/Pangu). Pangu is also a series of pre-trained AI models developed by Huawei Cloud that covers computer vision, natural language processing,

multimodal understanding, scientific computing (including weather forecasting) and so on.

## Data availability

For training and testing Pangu-Weather, we downloaded a subset of the ERA5 dataset (around 60 TB) from https://cds.climate.copernicus.eu/, the official website of Copernicus Climate Data (CDS). For comparison with operational IFS, we downloaded the forecast data and tropical cyclone tracking results of ECMWF from https://confluence.ecmwf.int/display/TIGGE, the official website of the TIGGE archive. We downloaded the ground-truth routes of tropical cyclones from the International Best Track Archive for Climate Stewardship (IBTrACS) project, https://www.ncei.noaa.gov/products/international-best-track-archive. All these data are publicly available for research purposes. Source data are provided with this paper.

## Code availability

The code base of Pangu-Weather was established on PyTorch, a Python-based library for deep learning. In building and optimizing the backbones, we made use of the code base of Swin transformer, available at https://github.com/microsoft/Swin-Transformer. Other details, including network architectures, modules, optimization tricks and hyperparameters, are available in the paper and the pseudocode. The computation of the CRPS metric relied on the xskillscore Python package, https://github.com/xarray-contrib/xskillscore/. The implementation of Perlin noise was inherited from a GitHub repository, https://github.com/pvigier/perlin-numpy. We also used other Python libraries, such as NumPy and Matplotlib, in the research project. We released the trained models, inference code and the pseudocode of details to the public at a GitHub repository: https://github.com/198808xc/Pangu-Weather (https://doi.org/10.5281/zenodo.7678849). The trained models allow the researchers to explore Pangu-Weather's ability on either ERA5 initial fields or ECMWF initial fields, where the latter is more practical as it can be used as an API for almost real-time weather forecasting.

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

**Acknowledgements** We thank ECMWF for offering the ERA5 dataset and the TIGGE archive; NOAA National Centers for Environmental Information for the IBTrACS dataset; and other members of the Pangu team for discussions and support with the GPUs. Our appreciation also goes to the Integration Verification team of Huawei Cloud EI, which offers us a platform of high-performance parallel computing.

**Author contributions** K.B. designed the project and trained the 3D deep networks for Pangu-Weather. L.X. improved the technical design. H.Z., X.C. and X.G. established the test environment and prepared for data. Q.T. managed and oversaw the research project. K.B. and L.X. wrote the paper.

**Competing interests** K.B., L.X., H.Z., X.C., X.G. and Q.T. are employees of Huawei Cloud. A provisional patent (not granted an ID yet) was filed covering the generative algorithm described in this paper, listing the authors K.B., L.X. and Q.T. as inventors.

**Additional information**
**Correspondence and requests for materials** should be addressed to Qi Tian.

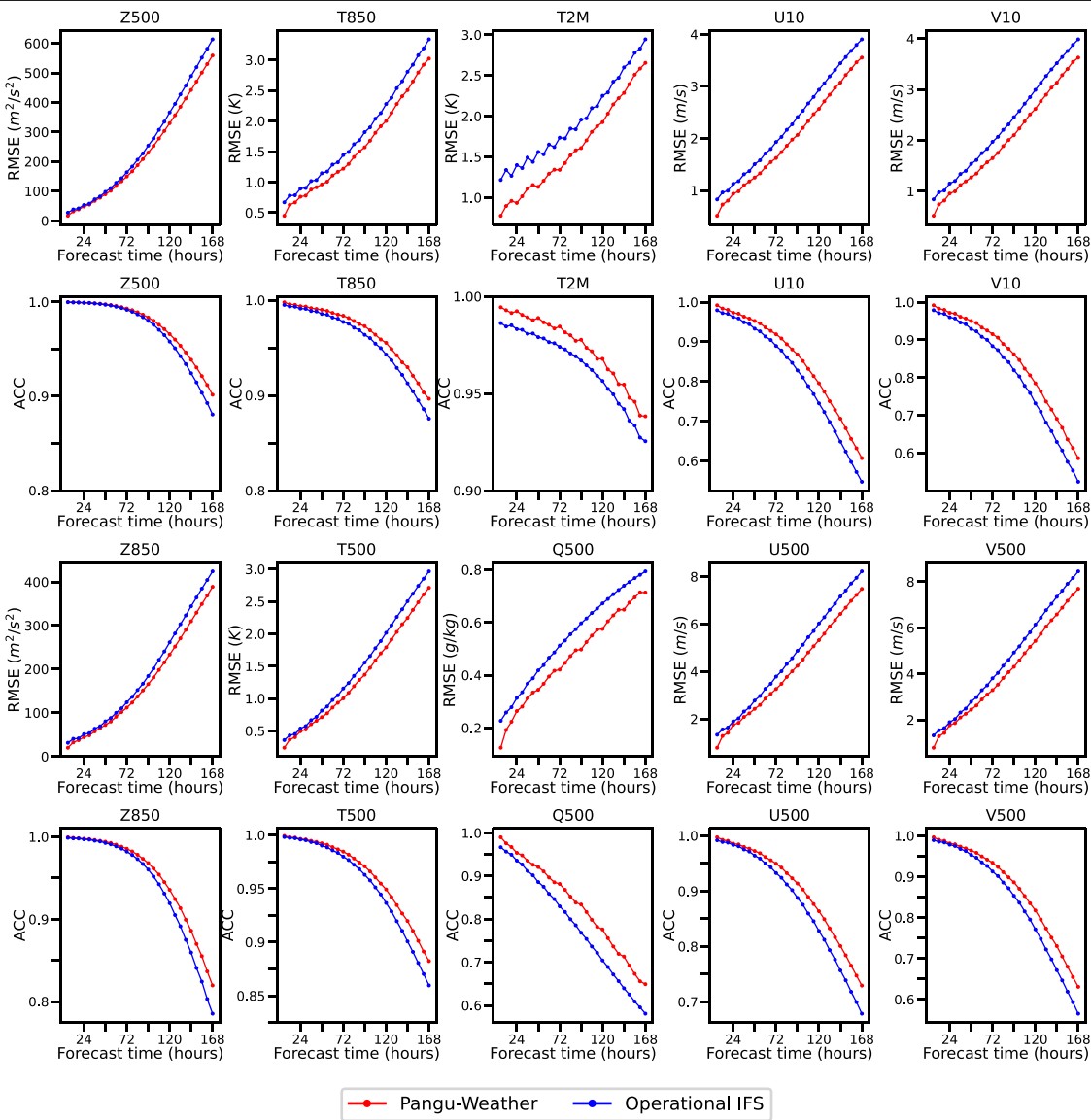

**Extended Data Fig. 1 | Deterministic forecast results in the Northern Hemisphere.** We only compared Pangu-Weather to operational IFS[3] because FourCastNet[2] did not report the breakdown results. We followed ECMWF to define the "Northern Hemisphere" to be the region between latitude of 20° (exclusive) and 90° (inclusive). Here, Z500/T500/Q500/U500/V500 indicates the geopotential, temperature, specific humidity, and $u$-component and $v$-component of wind speed at 500 hPa. Z850/T850 indicates the geopotential and temperature at 850 hPa. T2M indicates the 2 m temperature, and U10/V10 indicates the $u$-component and $v$-component of 10 m wind speed.

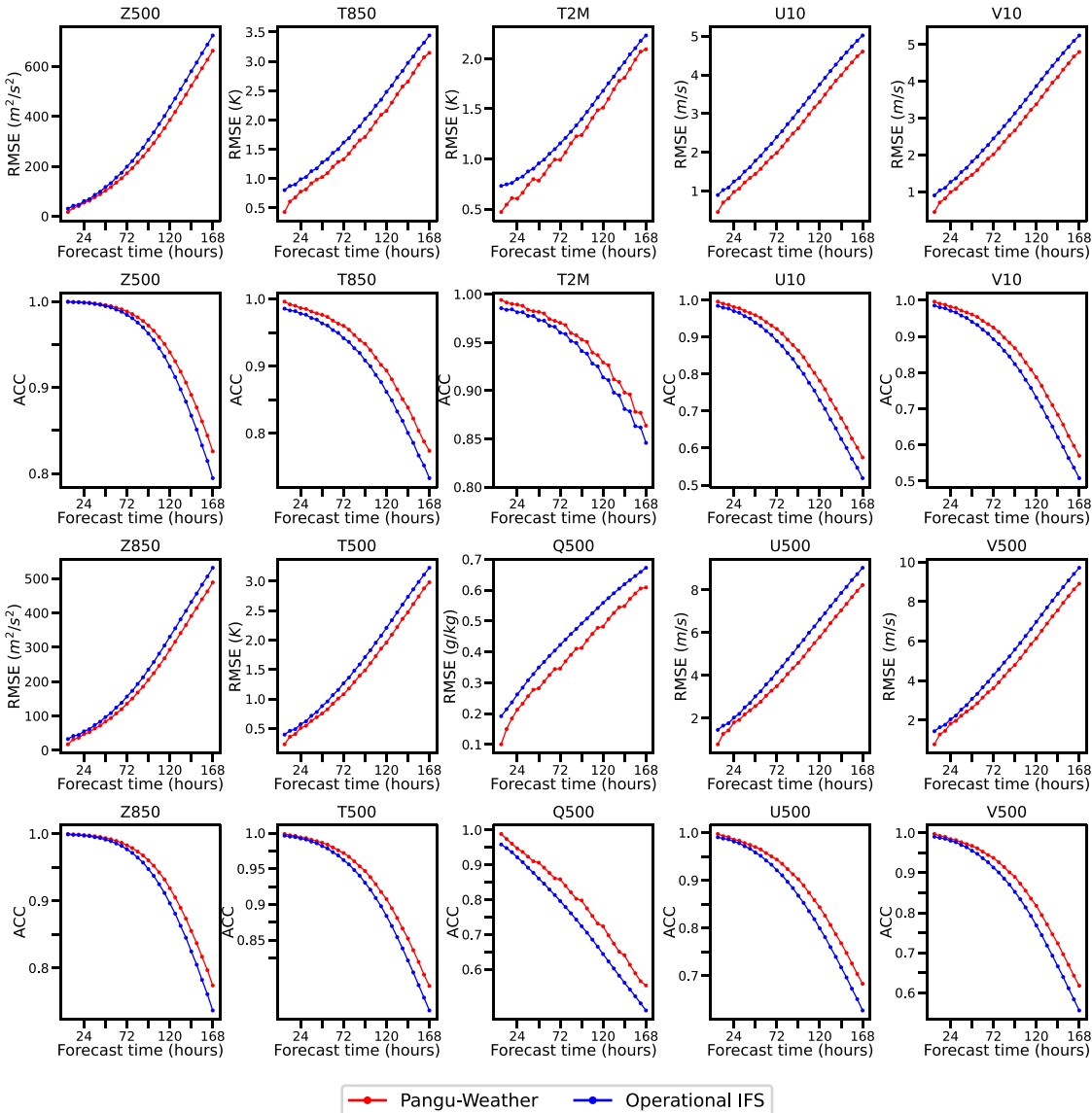

**Extended Data Fig. 2 | Deterministic forecast results in the Southern Hemisphere.** We only compared Pangu-Weather to operational IFS[3] because FourCastNet[2] did not report the breakdown results. We followed ECMWF to define the "Northern Hemisphere" to be the region between latitude of −20° (exclusive) and −90° (inclusive). Here, Z500/T500/Q500/U500/V500 indicates the geopotential, temperature, specific humidity, and $u$-component and $v$-component of wind speed at 500 hPa. Z850/T850 indicates the geopotential and temperature at 850 hPa. T2M indicates the 2 m temperature, and U10/V10 indicates the $u$-component and $v$-component of 10 m wind speed.

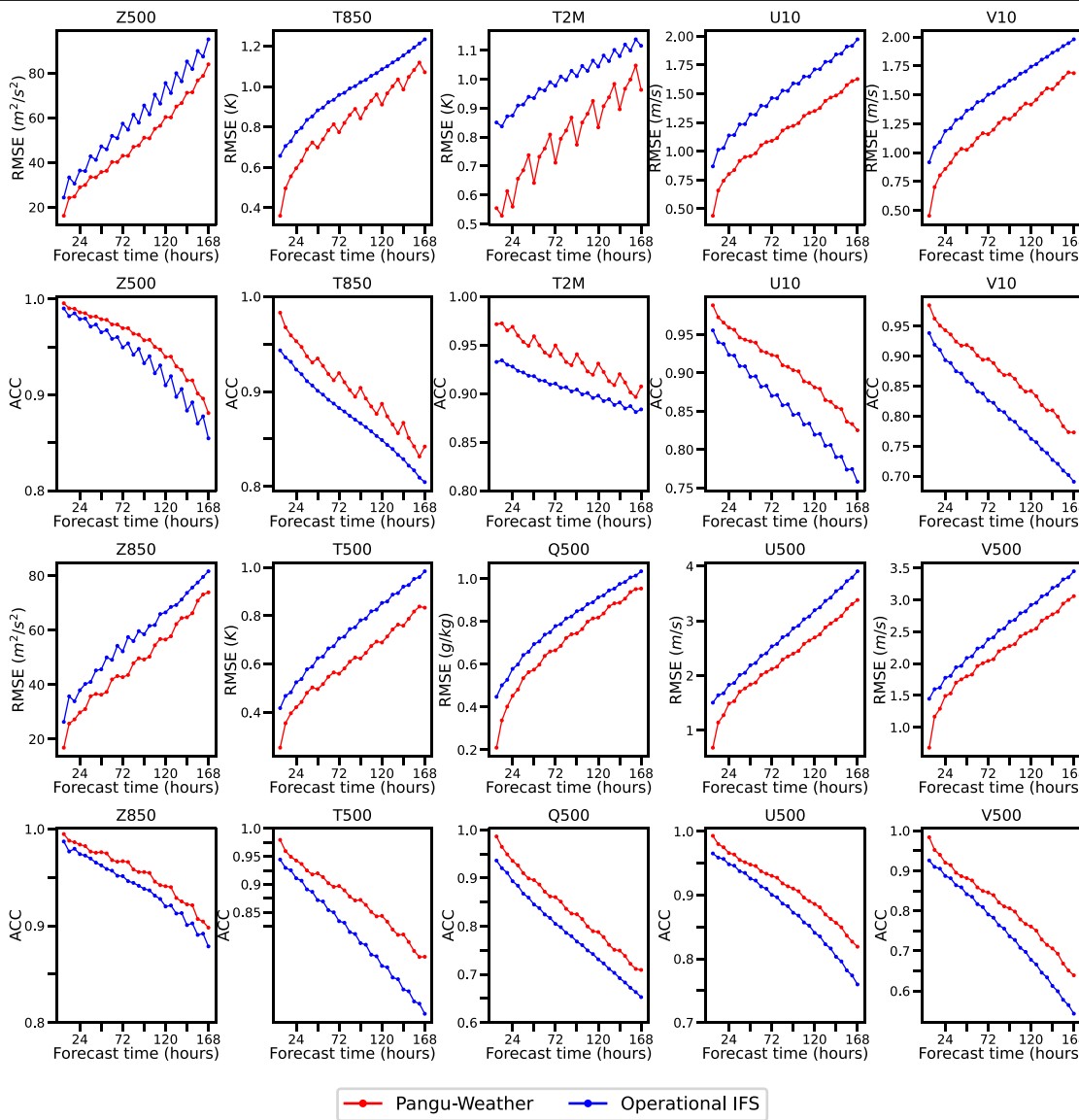

**Extended Data Fig. 3 | Deterministic forecast results in the tropics.** We only compared Pangu-Weather to operational IFS[3] because FourCastNet[2] did not report the breakdown results. We followed ECMWF to define the "tropics" to be the region between latitude of +20° (inclusive) and −20° (inclusive). Here, Z500/T500/Q500/U500/V500 indicates the geopotential, temperature, specific humidity, and $u$-component and $v$-component of wind speed at 500 hPa. Z850/T850 indicates the geopotential and temperature at 850 hPa. T2M indicates the 2 m temperature, and U10/V10 indicates the $u$-component and $v$-component of 10 m wind speed.

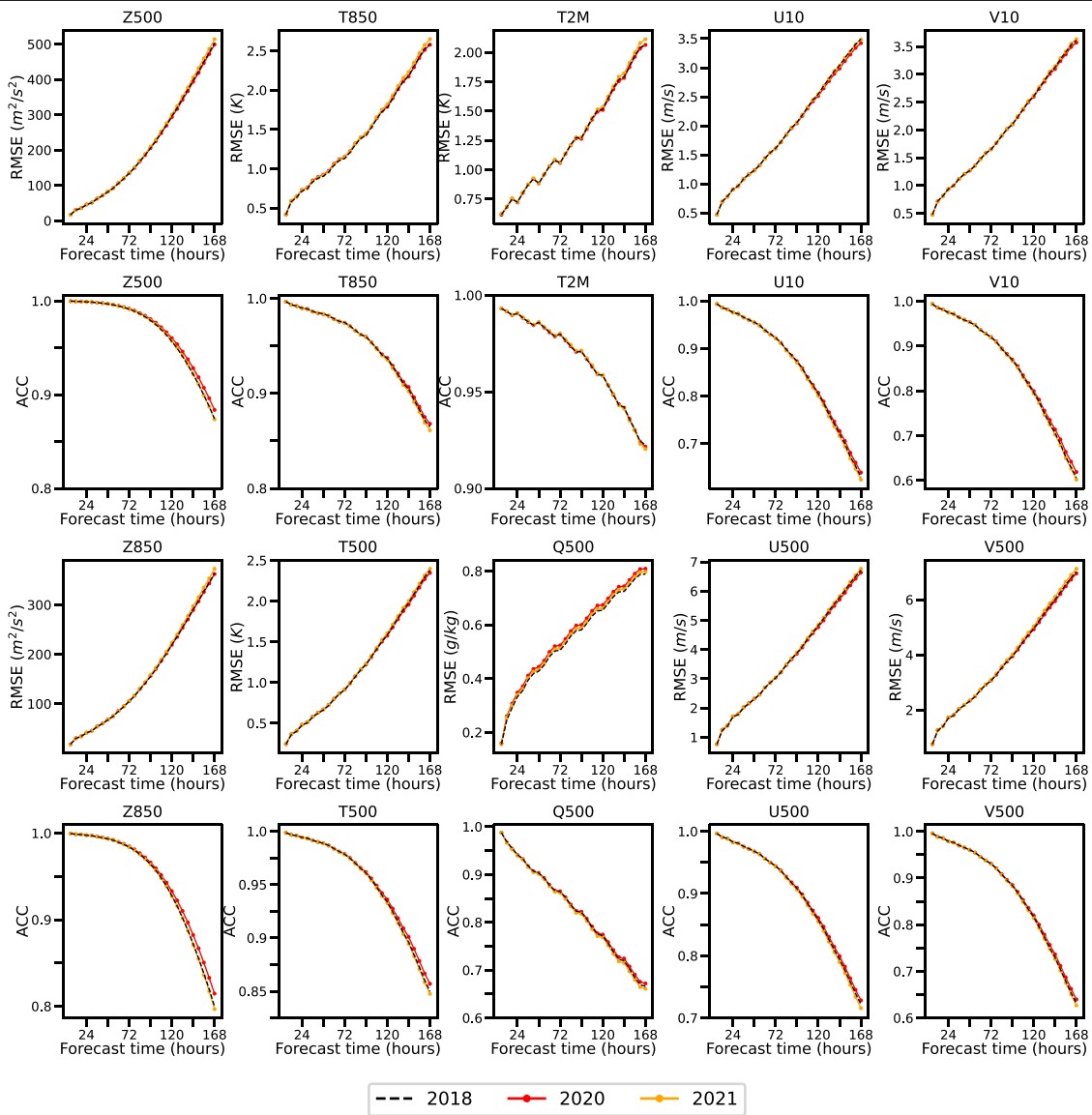

**Extended Data Fig. 4 | Deterministic forecast results of Pangu-Weather in 2018, 2020 and 2021.** The RMSE and ACC values and trends are close among the three years, indicating Pangu-Weather's stable forecasting skill over different years. Here, Z500/T500/Q500/U500/V500 indicates the geopotential, temperature, specific humidity, and *u*-component and *v*-component of wind speed at 500 hPa. Z850/T850 indicates the geopotential and temperature at 850 hPa. T2M indicates the 2 m temperature, and U10/V10 indicates the *u*-component and *v*-component of 10 m wind speed.

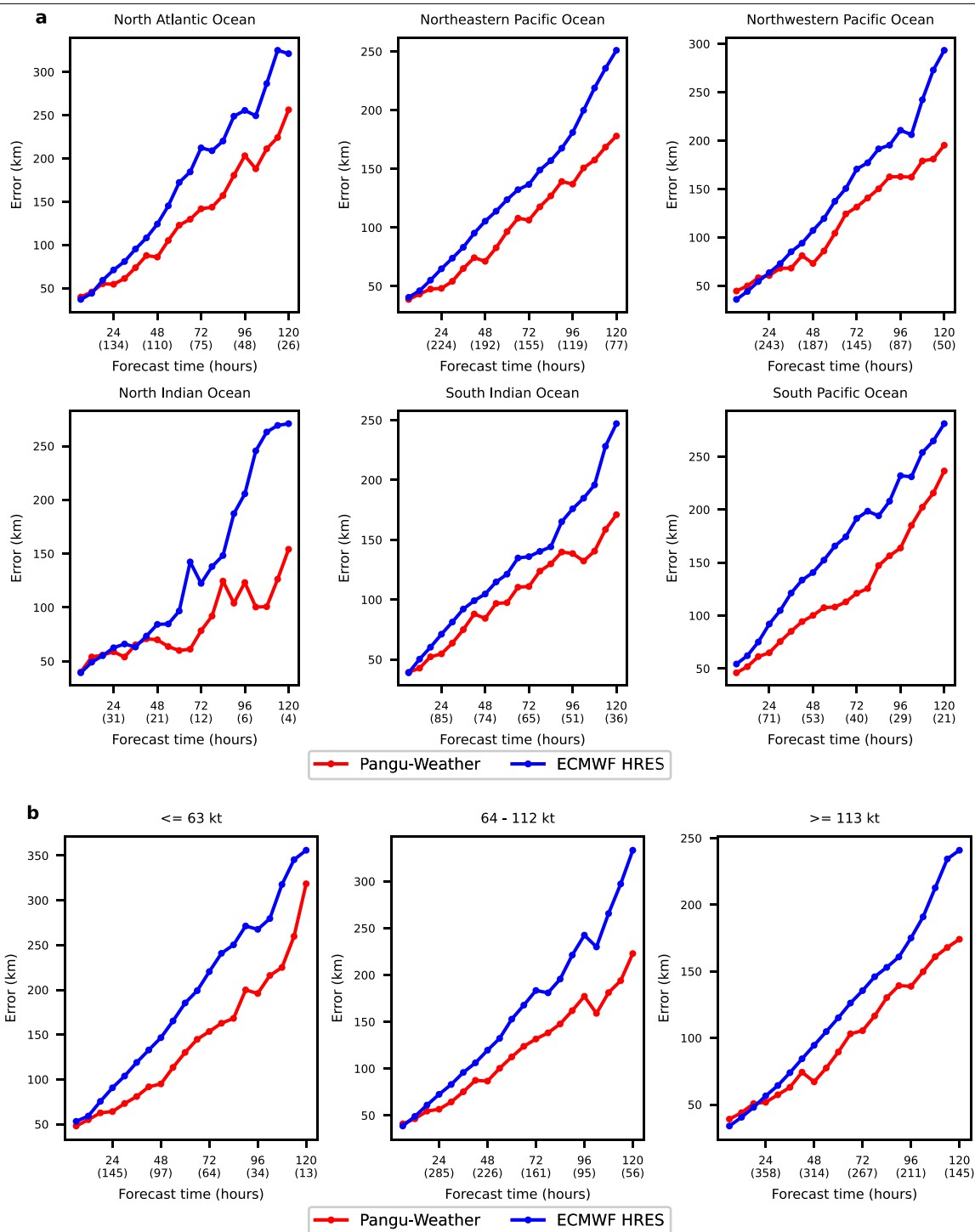

**Extended Data Fig. 5 | Breakdowns of the mean direct position errors of tracking tropical cyclones. a**) The breakdown into six oceans. **b**) The breakdown into three intensity intervals. The overall statistics is displayed in Fig. 4c.

**a**

Horizontal grids

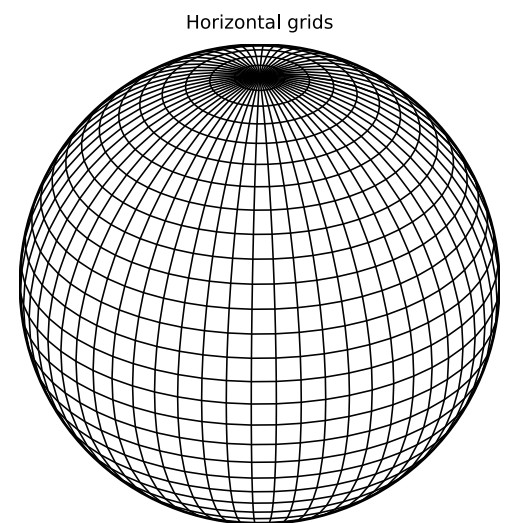

**b**

500hPa geopotential height

**c**

Distribution w.r.t. pressure level

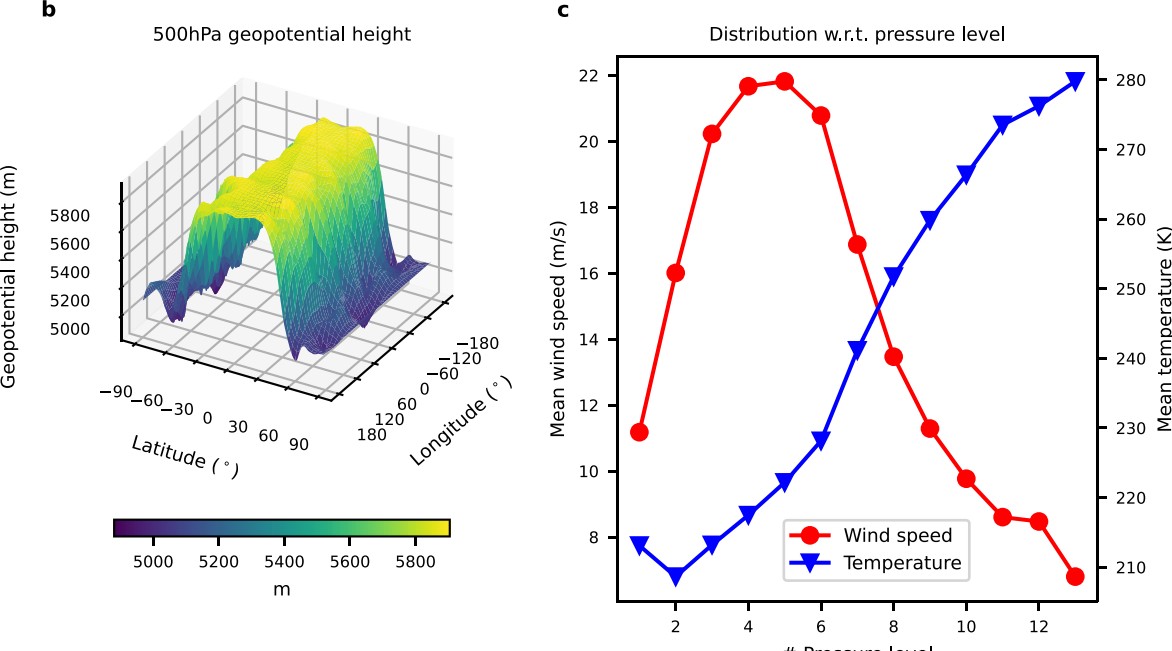

**Extended Data Fig. 6 | The motivation of using an Earth-specific positional bias. a**) The horizontal map corresponds to an uneven spatial distribution on Earth's sphere. **b**) The geopotential height is closely related to the latitude. **c**) The mean wind speed and temperature are closely related to the height (formulated as pressure levels). Sub-figures **b**) and **c**) were plotted using statistics on the ERA5 data.

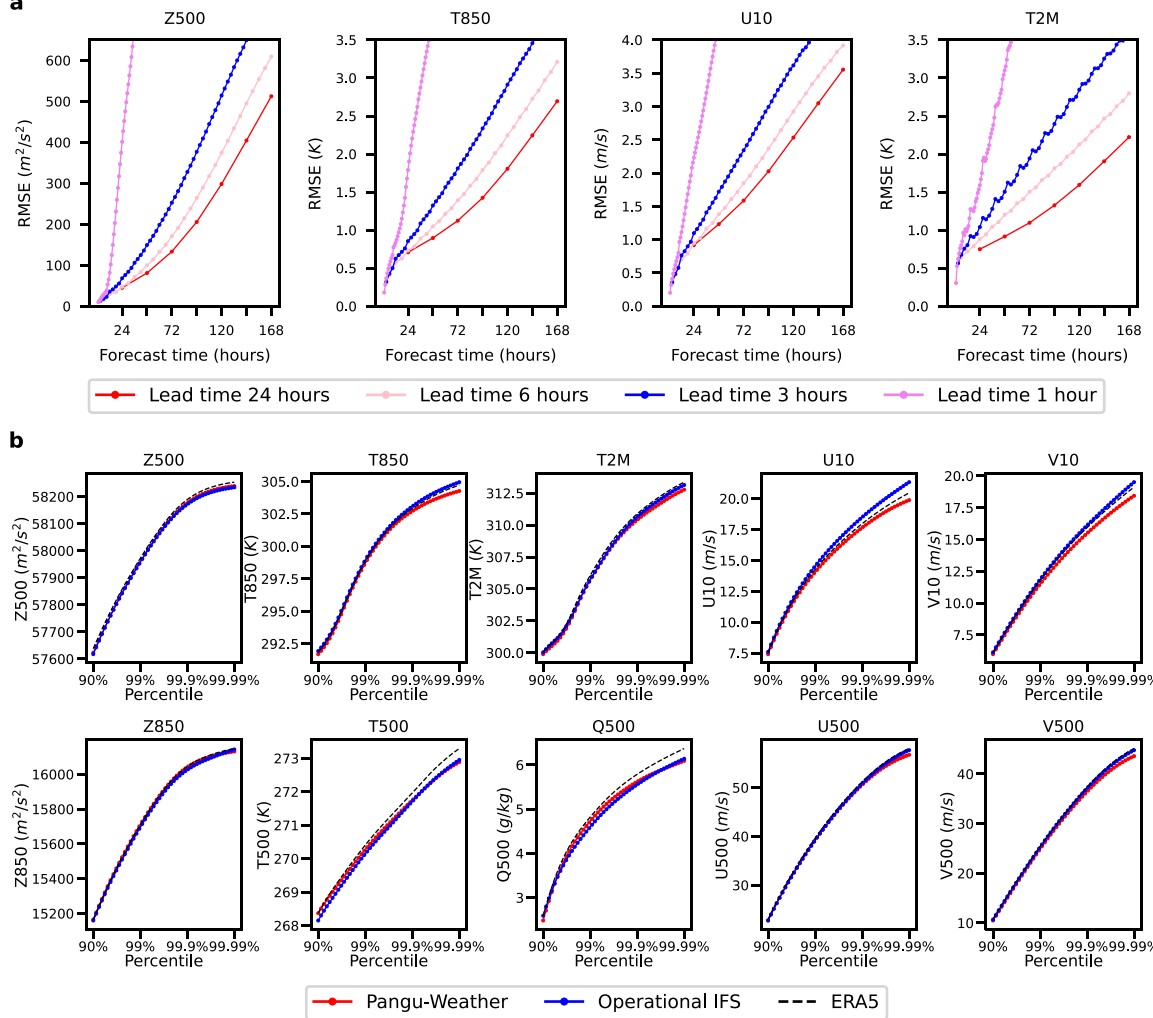

**Extended Data Fig. 7 | Properties of deterministic forecast results.**
**a)** Single-model test errors. It shows the test errors (in RMSE) with respect
to forecast time using single models (i.e., lead times being 1 h, 3 h, 6 h, and
24 h, respectively). Mind the accumulation of forecast errors as forecast time
increases. **b)** Visualization of the trend of quantiles with respect to lead time.
It shows the trend of all the variables displayed in Fig. 2 and the comparisons
to operational IFS[3] and ERA5[18]. Pangu-Weather often reports lower quantile

values because AI-based methods tend to produce smooth forecasts. Here,
Z500/T500/Q500/U500/V500 indicates the geopotential, temperature,
specific humidity, and $u$-component and $v$-component of wind speed at
500 hPa. Z850/T850 indicates the geopotential and temperature at 850 hPa.
T2M indicates the 2 m temperature, and U10/V10 indicates the $u$-component
and $v$-component of 10 m wind speed.

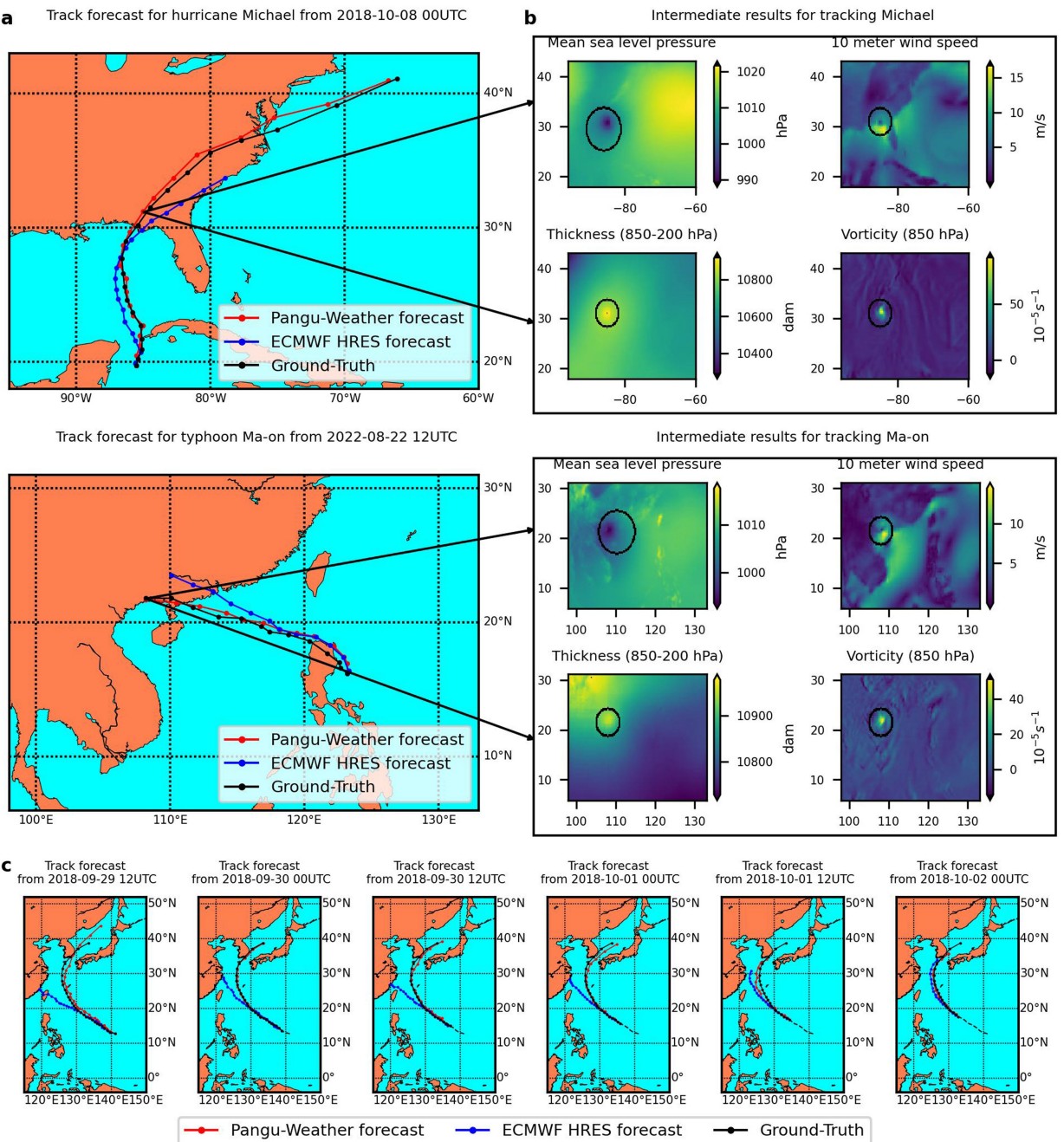

**Extended Data Fig. 8 | Visualization of tracking tropical cyclones. a)** The tracking results of cyclone eyes for Hurricane Michael (2018–13) and Typhoon Ma-on (2022–09) by Pangu-Weather and ECMWF-HRES, with a comparison to the ground-truth (by IBTrACS[24,25]). **b)** An illustration of the tracking process, where we used Pangu-Weather as an example. The algorithm locates the cyclone eye by checking four variables (from forecast results), namely, mean sea level pressure, 10 m wind speed, the thickness between 850 hPa and 200 hPa, and the vorticity of 850 hPa. The displayed figures correspond to the forecast results of these variables at a lead time of 72 h, and the tracked cyclone eyes are indicated using the tail of arrows. **c)** The procedural tracking results of Typhoon Kong-rey (2018–25). The results of Pangu-Weather were compared to that of ECMWF-HRES and the ground-truth (by IBTrACS[24,25]). We show six time points with the first one being 12:00 UTC, September 29th, 2018, and the time gap between neighboring sub-figures being 12 h. The historical (observed) path of cyclone eyes is shown in dashed. Mind the significant difference between the tracking results of Pangu-Weather and ECMWF-HRES (Pangu-Weather is more accurate) at the middle four sub-figures. The sub-figures with maps were plotted using the Matplotlib Basemap toolkit.

**Extended Data Table 1 | The correspondence of upper-air and surface variable names and their abbreviations**

| Type | Full name | Abbreviation |
|---|---|---|
| Upper-air variables | geopotential | Z |
| | specific humidity | Q |
| | temperature | T |
| | $u$-component of wind speed | U |
| | $v$-component of wind speed | V |
| Surface variables | mean sea level pressure | MSLP |
| | 2m temperature | T2M |
| | $u$-component of 10m wind speed | U10 |
| | $v$-component of 10m wind speed | V10 |

Throughout the paper, we extracted the upper-air variables from 13 out of 37 pressure levels (50 hPa, 100 hPa, 150 hPa, 200 hPa, 250 hPa, 300 hPa, 400 hPa, 500 hPa, 600 hPa, 700 hPa, 850 hPa, 925 hPa, 1000 hPa) plus the surface variables. Therefore, a total of 69 variables were used as inputs.

**Extended Data Table 2 | The forecast time gain of Pangu-Weather**

| Variable | Gain over operational IFS (h) | Gain over FourCastNet (h) |
|---|---|---|
| Z500 | 10.45 | 43.23 |
| T850 | 15.37 | 41.05 |
| T2M | 18.19 | 43.11 |
| U10 | 19.68 | 43.81 |
| V10 | 19.10 | 42.78 |
| Z850 | 10.62 | N/A |
| T500 | 13.66 | N/A |
| Q500 | 31.00 | N/A |
| U500 | 17.52 | N/A |
| V500 | 16.16 | N/A |

It lists the 10 weather variables shown in Fig. 1, namely, Z500/T850/T2M/U10/V10 on which both operational IFS[3] and FourCastNet[2] reported, and Z850/T500/Q500/U500/V500 on which only operational IFS reported. Here, Z500/T500/Q500/U500/V500 indicates the geopotential, temperature, specific humidity, and $u$-component and $v$-component of wind speed at 500 hPa. Z850/T850 indicates the geopotential and temperature at 850 hPa. T2M indicates the 2 m temperature, and U10/V10 indicates the $u$-component and $v$-component of 10 m wind speed. To calculate the values for a given variable, we first fetched the RMSE value reported by Pangu-Weather in a 168-hour forecast, and then estimated the forecast time gain, denoted by $\Delta t$, so that the compared algorithm (e.g. operational IFS) report the same RMSE value when the lead time is 168 h minus $\Delta t$.