## [Peer Review File · Nature]

Manuscript Title: Accurate medium-range global weather forecasting with 3D neural networks

Reviewer Comments & Author Rebuttals

Reviewer Reports on the Initial Version:

Referees' comments:

Referee #1 (Remarks to the Author):

Pangu Weather presents a novel deep learning model for accurate weather forecasts, which surpasses, or at least comes close to the forecasting skills of one of the best numerical weather prediction models. It is one of three recent breakthroughs in the application of deep learning to weather forecasting and an extension of the Swin transformer model, which has shown some advantages over other approaches in the field of image analysis.

Overall, the paper presents robust results and provides a clear description of the methodology and data. However, as pointed out in my major concern comment below, the assessment of the model quality is somewhat overstated. If these exaggerations are removed from the paper, it becomes a very valuable contribution to the field.

Major concern:

The description of input data sampling suggests that you have applied random sampling of the 340 k samples of 39 years of ERA5 data, and indeed this is confirmed in the detailed description in the annex material. This leads to over optimistic evaluation results due to auto correlation of weather data (see discussion in Schultz et al., 2021). Furthermore, Pangu Weather was trained on ERA5 data with updated analyses from observations at least every 24 hours. Therefore, any forecasting evaluation presented in the paper is effectively an interpolation, whereas ECMWF-HRES results presented as comparison are true forecasts, i.e. the model never had any knowledge about the future weather. It is therefore questionable, whether Pangu Weather really performs better than IFS or even FourCastNet for real weather forecast as claimed in lines 117-131 of the manuscript and shown in Figure 2. While the results are still impressive, the manuscript needs to downtone its claims, because a more careful evaluation is needed before it can truly be said that Pangu Weather outperforms "the world's best NWP model".

General comment:

The language needs polishing by a native speaker. For example, single case and plural are sometimes mixed in one sentence. While I corrected some language errors, I did not pay full attention to these in all parts of the manuscript.

Minor comments:

l.7 "formulate the physical rules of atmospheric states" is a rather sloppy formulation. The PDEs rather describe the transition between atmospheric states. Also worthwhile to mention the

discretisation of NWP models.

I.20 Please downtone the conclusion a little bit. There are still a few steps to be taken, before AI models are really better than NWP. In particular, as mentioned above, Pangu Weather needs to be run in a true forecast setting with input data it has never seen during training. Also, NWP models make direct use of observational data, while Pangu Weather requires a NWP model field to issue a forecast. Rather than "progress of AI surpassing NWP" it should say something like "progress on the path of establishing AI as a complement or surrogate of NWP" - the remark that it was thought to be far away should be moved to the Introduction and removed from the abstract, because it is still a bit away.

I.22 wording: "application" instead of "scenario"

I.24 "high performance computers" instead of "the ... device"

I.28 imprecise wording "parameterisations", not "parametric models"; it should be said that parameterisations are needed to capture subgrid-scale (or unresolved) processes. As I suggest to delete the final discussion on NWP versus AI-based models in the annex, you could re-use some of the information from there to state more clearly what empirical parameterisations are.

I.33 equating "input" with "observed data" gives a wrong impression that you were using observations as inputs, while in fact this was reanalysis data, hence, ultimately, a (NWP) model result.

I.47 this could be confusing to readers not familiar with the AI literature: if 3D "can better capture the relationship between pressure levels", then one may wonder "compared to what baseline?". Have earlier models not had more than one level? Or did they have no control over the coupling?

I.60 I wouldn't call 297 "much" lower than 333. Please delete "much"

I.62 why provide the time in ms when other values are given in s or h? Replace with "1.4 s".

I.63 "sufficient details": sufficient for what purpose. Clearly, it can be seen that PanguWeather forecasts are more blurred than ERA5 fields. It might be worth explaining the "bluriness" a bit more: the cyclone tracking images in the annex suggest that, at least for some fields, Pangu Weather results are not smoothed at all.

I.65 "cyclone tracking" is an application, not a scenario

I.66 Be careful with the claim that PanguWeather is the first DL model that can accomplish cyclone tracking. See, for example, <https://www.frontiersin.org/articles/10.3389/fdata.2020.00001/full> and <https://gmd.copernicus.org/articles/14/107/2021/gmd-14-107-2021.html> .

I. 69 please use less "dramatic" language. It would be more interesting to readers to get to know in which way the ECMWF tracking failed. Was it the typhoon path, its intensity, the timing?

I. 72 it is technically not fully correct to refer to the ERA5 input data as "observations". Substantial processing including error corrections went into observations and the IFS model acts as a sort of filter (i.e. data assimilation). Thus, while strongly bound by observations, the input data you have used are not directly observations.

I. 74-77 This description looks like you applied random sampling of individual hourly values. Schultz et al. (2021), your reference #4, discussed the problems with this method. Basically, random sampling makes your model evaluation less meaningful due to auto correlation in the data.

I. 88 What exactly do you mean with "Earth-specific positional bias"? Please add a reference to the extra material, where a description is given.

I. 81-91 and Figure 1 The network architecture is interesting and overall well described. However, it may be useful to add another panel to Figure 1, or modify panel b, to document more clearly, how the 4 networks interact, if at all. From the panel b it appears that they run completely independently, but this cannot be, because the 54th hour in your example, of course depends on the first two iterations of the 24-hour model, i.e. there must be some sequential coupling. You do mention this implicitly in I. 103, but please make it more explicit.

I. 140-147 This is a somewhat awkward discussion about the model smoothness. It is a general property of any regression algorithm (and a large transformer network is nothing else) to converge to average values. In contrast, a single IFS forecast calculates a single estimated value at each grid cell and for each time step by solving a system of PDEs with initial conditions. Due to the chaotic nature of weather and the inevitably imprecise knowledge of the initial conditions and subgrid scale processes, weather forecasters have long known that there are statistical uncertainties associated with each forecast, and they have thus introduced the concept of ensemble forecasting. Overly smooth fields can never be seen as advantage.

I. 155 This statement doesn't make sense: extreme events are characterized as being at the tails (usually upper tail) of a variable distribution. Their relevance stems from the fact that they are extreme, and "accuracy" can only refer to getting the magnitude right. It can make an enormous difference for people whether a rain event generates 80 or 120 mmm of rain. Of course, there are other relevant evaluation criteria for extreme events, such as the correct timing and advection path. These must be discussed separately.

I. 158 "specific treatment of extreme weather data" is too unspecific. This remains one of the most difficult challenges for applying AI trustfully to weather data. Unless you document some ideas, this statement should be better formulated as a remaining challenge rather than an expectation.

I. 160-173 The typhoon tracking results are impressive. However, as stated in my major comment, the comparison is somewhat unfair, because the ECMWF HRES model had no knowledge about the future, whereas Pangu Weather was trained on data with mor e frequent input from observations, and the tracking results have not been generated with completely independent input data, i.e. data that has never been seen during training or fine-tuning.

I. 172ff Please add more specific information to the Perlin noise perturbation that was used to generate the ensemble forecasts. For example: what was the maximum length of the noise vector? How were vector lengths distributed?

I. 205 As stated above, the claim that "Pangu-Weather, for the first time, surpasses world's best NWP system" is over stated.

I. 214 What do you mean by "observation factors"? Are these additional variables on more vertical levels?

I. 289 please insert a reference to the "extended data table 1" in the annex.

I. 295 The evaluation remains a bit shallow if compared to the established standards in numerical weather modeling. See for example <https://sites.ecmwf.int/ifs/scorecards/scorecards-47r3HRES.html> for a regional breakdown of results. At a minimum, I suggest that the metrics are evaluated also for Northern and Southern hemisphere separately. It would also be highly beneficial to get a separate metric for the tropics, because this is the region where classical numerical models have the largest uncertainty.

I. 305ff Here, another citation of Hersbach et al., (2020) would be appropriate.

I. 355 Please include a reference to Swin transformer here, because the window shift is a key feature of Swin T.

I. 356 "Along the longitude ..." this sentence is incomprehensible.

I. 367 delete "modify"

I. 371 the "neurons" are not partitioned; please use the term "data volume" or similar instead.

I. 376 confusing: the window coordinate should be only (mpl, mlat) if mlon is not used. You could also write "2-dimensional window coordinate".

I. 380 Even though it is understandable that you cannot fully train a large number of network configurations, it would nevertheless be illustrative to show loss curves or accuracy metrics of at least some variants over a couple of training epochs and discuss the observed differences.

I. 390 Is there any evidence to support your believes that 4D tensors would lead to better results? Have you run tests with smaller configurations? Is there a paper in the literature that proves this for other types of data?

I. 401 I suggest to delete "while still being one of the fastest systems...". It is sufficient to say that Pangu Weather is about 10.000 faster than IFS and of comparable speed as Fourcastnet.

I. 406 "follow" instead if "following"

I. 404 ff The RQE does not evaluate if extreme values occur at the right location and time, but only looks at the tails of the value distribution. While the typhoon tracking results indicate that Pangu Weather can also capture individual extreme events, it would be good to show this more explicitly.

I. 432 This discussion of cyclone tracking results is unfair, because ECMWF-HRES had no information about the future when it had to issue the forecasts, whereas Pangu Weather was trained on ERA5 data with daily updated analyses. Furthermore, as no data was held back for the evaluation, Pangu Weather "only" had to interpolate between data it has seen during training. While the results are still very good, this is not a final verdict of the true forecasting quality of Pangu Weather. For this, you should evaluate cyclones in a year that was not used for training, e.g. 2022.

I. 467 what do you mean by "good practice"?

I. 468 ff This summary of previous work is somewhat awkward. The issues with NWP models have been discussed in other papers, and the benefits or approaches of AI-based methods are not explained well enough to convince readers. What is required to be known about the AI approach and specifically Pangu Weather is (and should be) contained in the main text and the detailed network and data descriptions above. I don't think, this section is necessary and the manuscript would be better without it.

I. 522 "code base *of* ..."

I. 597 (extended data table 1): please add the levels from which data were extracted in the table caption and the total number of resulting variables so that all relevant information is accessible at one place.

Code and data availability: I could trace back the data sources and ran the code provided at the link that was made available to the reviewers. This worked well, and the code is well structured. The only comment there would be to collect the library requirements in a requirements.txt file instead of listing them in the README. Furthermore, the authors should of course release their code publicly when the paper is accepted and before publication.

Referee #2 (Remarks to the Author):

In this manuscript the authors tackle medium range weather forecasting using machine learning. They develop a 3D transformer model with the ability to learn location specific terms and train on the ERA5 archive. The model shows strong results for both headline scores and tropical cyclones, indicating that a useful tool has been learnt.

The task itself is not novel, but the model has novelty. The results themselves are a significant step beyond previous results. This work will, in my opinion, make people reevaluate what forecasting models might look like in the future.

On data and methodology, the data used for training and evaluation is appropriate. There are two issues with the analysis methodology. The section on ensemble forecasting completely ignores a key point, that ensembles should be assessed through probabilistic scores, not deterministic ones. Figure 5 should be replaced. My suggestion is that the RMSE plot be replaced by a CRPS plot (which is defined in equation 4 here <https://arxiv.org/pdf/2205.00865.pdf>), this could still compare the deterministic and ensemble models, as the CRPS for a deterministic forecast reduces to the MAE. This would be a valuable addition to the community as future projects can then compare against Pangu's results. I would additionally suggest the authors plot the spread/error ratio (defined in the same paper as equations 1 & 3) as a function of lead time. This shows the reliability of the forecast. Secondly, RQE as a single number lacks interpretability and does not add value. This data would be best shown by showing the humidity value distribution rather than aggregating to a single number, for example a plot similar to the left panels in FourCastNet's figure 9. This should be shown for a lead time such as 3 or 5 days, perhaps both.

The broad conclusions are appropriate but they ignore several important points which I think should be added. While, I agree with the authors that there is room for improvement on the AI side. I think it is worth noting that there is also room for improvement on the conventional modelling side. NWP models develop biases which are predictable and therefore can be compensated for with post-processing of the NWP output. In the search for the optimal forecasting model it is still possible that a combination of NWP and ML-based post-processing produces the optimal product. I would like to see this noted in the conclusions. Perhaps most importantly, ERA5, whilst of high quality is not a universal truth. A true test of any model would be against quality-controlled observations. These are as easily accessible as ERA5, so suggesting a comparison is not within the scope of the paper. However this qualification should be stated at key moments in the paper. For example the abstract should read
"...surpasses NWP methods in terms of forecast accuracy when measured against reanalysis." In the discussion this comparison against observations should be highlighted as vital future work to further establish the skill of Pangu-Weather.

The references are in general correct, although on L149 I believe this citation is incorrect and should be for FourCastNet.

Other comments:

In the authors' manuscript on a public archive system they included a breakdown of TC track errors by region and intensity. I found this figure most interesting and would suggest it is added to the appendix of this work.

"ECMWF-HRES, world's best tracking system, failed dramatically" -- this is an inaccurate representation of these events. TC tracks for both of these events were well-predicted, however not with as early of a warning as Pangu demonstrates. Both in the text and caption it should be noted that Pangu identifies the correct track earlier, rather than suggesting that the events were forecast busts for the IFS.

Temporal consistency is an important feature of weather forecasts, for example rainfall over many

hours will contribute to flooding (understanding that rainfall is not an output of PanguWeather). How does Pangu weather, by using multiple networks, ensure temporal consistency? For example the predictions for 23 and 24 hours are the product of different models/steps. I would label this an outstanding challenge in your conclusions.

L124/5 It would be useful to have a table showing the "forecast gain time" of Pangu vs the baseline methods across the various variables.

L140 "This is a typical property of deep neural networks in learning from large-scale datasets" -- it should be noted that smooth predictions for some fields, particularly those associated with moist processes, can degrade the impact of forecasts, for example less extreme rainfall events than conventional models.

L144 "Still, Pangu-Weather produces high-resolution forecasts that are very close to the ground-truth and preserve most of small-scale structures of surface variables." -- how are you measuring or assessing this?

L213 "(1) incorporating more observation factors" I don't understand this point. Please could you reformulate?

Minor comments:

L9 potentials -> potential

Referee #3 (Remarks to the Author):

A) Summary:

This manuscript presents an AI model for medium-range global weather forecasting, Pangu-weather. The model is a neural network that is trained on temporal sequences of global ERA5 data. It is completely data-driven, i.e. there are no physical equations used to derive the model, only ERA5 data. The model is part of a new hot trend that seeks to leverage the newest advances in deep learning, specifically the powerful new "transformer" architecture, to create AI models to tackle this difficult problem. The first big milestone along those lines was FourCastNet - released by NVIDIA in Feb 2022 – that predicted various atmospheric variables at several different pressure levels. The model presented here is the second major milestone which seems to exceed the abilities of FourCastNet in two aspects: 1) it models the atmosphere as 3D (with very coarse vertical discretization), rather than as several disconnected 2D slices as FourCastNet does; 2) it appears to provide significantly higher accuracy than FourCastNet.

B) Novelty/Originality:

Level of novelty/originality: high (A+)

The proposed neural network architecture to model the atmosphere in 3D, rather than 2D slices, is novel and seems to work well.

Level of importance of results: high (A+)

C) Data & methodology:

Methodology: I have to admit that I did not follow all the details 100%, since it builds on other work I have not yet read, specifically on “Swin transformers”. But their model choices appear to be clean and logical – not a hack, but instead building on well established methods from other areas and adjusting them to the purposes of this application.

Data: Please see Major Concern #3 below.

D) Statistics/uncertainty

Uncertainty is not analyzed here, which is ok. Error statistics provided are a good first start.

E) Conclusions: There are several limitations that you did not discuss, see Concerns #1-#3 below.

F) Suggested improvements:

Major Concerns:

Overall, I applaud the team for lots of progress. This work represents a significant step forward in the development of data-driven global weather prediction models. Yet there are also major shortcomings that I would like them see address. #1 - #3 below can probably be addressed by discussing these limitations. But #4 is the one that should be addressed.

1) Your weather forecast model does not predict precipitation! Precipitation is a crucial part of weather forecasts. It seems odd to evaluate a global weather forecasting system that does not even try to provide precipitation forecasts.

2) Not predicting precipitation also implies that the forecasts of your other variables cannot utilize precipitation estimates at prior time steps. I would expect the lack of precipitation data to cause problems for the accurate prediction of small-scale extreme weather events (such as tornado outbreaks, see <https://doi.org/10.1175/MWR-D-21-0013.1> , <https://doi.org/10.1175/MWR-D-21-0014.1>). That might be why both FourCastNet and Pangu-Weather focus on case studies for large-scale extreme events, such as hurricanes. I don't expect you to fix this problem, but at least discuss this limitation, please.

3) You only provide results using ERA5 data, which is not available in real-time. How well does your model work when you feed in real-time (e.g. GFS) data instead?

4) Reproducibility / advancing science:

The pseudo-code provided with the paper is a great resource for anyone who wants to implement this model, as it provides many more details than the standard schematics provided in most papers. Well done. However, implementing the model would still require significant expertise and effort from an expert AI team. More importantly, training the model requires significant GPU resources, more than most academic groups are likely to have access, too. Training took 16h each for the four

required models on a 192 GPU system – not something that most research groups have access to.

What would truly accelerate science would be to provide – in addition to the pseudo code - a fully trained model that research groups can download and run locally, such as NVIDIA provided for FourCastNet, see <https://github.com/NVlabs/FourCastNet>. This allows researchers to run such a model, in seconds, on a local machine with just one GPU. Providing Pangu-weather in a similar format would allow researchers – even those with few resources and little AI knowledge - to test such a model to their heart’s desire. That would allow the meteorological community to evaluate the strengths and weaknesses of such models, to provide feedback on how to improve it, and thus speed up the development of models that will be useful for society. Will you accept the challenge to make your model available in a similar form?

Minor issues:

1) It would be fun to include a paragraph on why this model is named “Pangu”. I assume it’s named after the Chinese mythology figure “Pangu” (<https://en.wikipedia.org/wiki/Pangu>)? Would be great to provide a bit of cultural context/education to the audience.

2) The English in the main document needs some polishing – the meaning is generally clear but the English is clumsy.

Example: high-performance computational device \diamond high-performance computing system.

The methods part is fine as is.

G) References are good. Extensive comparison to prior model, FourCastNet, is excellent.

H) Clarity and context: very good.

Author Rebuttals to Initial Comments:

Response to Referee #1

Q1: Pangu Weather presents a novel deep learning model for accurate weather forecasts, which surpasses, or at least comes close to the forecasting skills of one of the best numerical weather prediction models. It is one of three recent breakthroughs in the application of deep learning to weather forecasting and an extension of the Swin transformer model, which has shown some advantages over other approaches in the field of image analysis.

A1: Thanks for the summary and agreeing with our contribution.

Q2: Overall, the paper presents robust results and provides a clear description of the methodology and data. However, as pointed out in my major concern comment below, the assessment of the model quality is somewhat overstated. If these exaggerations are removed from the paper, it becomes a very valuable contribution to the field.

A2: Thanks for the kind reminder. We followed your suggestion to remove the exaggerations from the manuscript to avoid over-claiming our contribution and our model's quality. Specifically,

- We avoided claiming that “Pangu-Weather surpasses operational IFS” and changed the statement into something like “Pangu-Weather obtains better deterministic forecast results on reanalysis data compared to operational IFS.”
- We changed the improper statements “... that even ECMWF-HRES, world's best tracking system, failed dramatically” into “... that remain a challenge for the world's best tracking systems such as ECMWF-HRES.”
- We added a paragraph to the end of the “Discussions” part, trying our best to discuss the current limitations of AI-based methods for weather forecast. In the next paragraph, we pointed out that combining the ability of NWP and AI-based methods is a future research direction.

Q3: The description of input data sampling suggests that you have applied random sampling of the 340 k samples of 39 years of ERA5 data, and indeed this is confirmed in the detailed description in the annex material. This leads to over optimistic evaluation results due to auto correlation of weather data (see discussion in Schultz et al., 2021). Furthermore, Pangu Weather was trained on ERA5 data with updated analyses from observations at least every 24 hours. Therefore, any forecasting evaluation presented in the paper is effectively an interpolation, whereas ECMWF-HRES results presented as comparison are true forecasts, i.e. the model never had any knowledge about the future weather. It is therefore questionable, whether Pangu Weather really performs better than IFS or even FourCastNet for real weather forecast as claimed in lines 117-131 of the manuscript and shown in Figure 2. While the results are still impressive, the manuscript needs to downtone its claims, because a more careful evaluation is needed before it can truly be said that Pangu Weather outperforms “the world's best NWP model”.

A3: There are several sub-questions here, and we respond to them separately.

(1) There might be some misunderstandings here. As claimed in L.60, the training set is composed of the reanalysis data in 1979-2017, the validation set in 2019, and the testing set in 2018, 2020, 2021. This is similar to Case (b) in Figure 4 of (Schultz et al., 2021). That said, our algorithm has seen no training data

on or after 2018 when tested on the 2018 data. By “we sample the input time point from the training set” (original version: L.75) we mean to randomly shuffle the order of data points to reduce the risk that the deep learning models over-fit on training data. To be clear. Let the training set contains 340K data points, numbered as (1,2,...,340K) by starting time. When random shuffle is not performed, in each epoch, the model sees these data points in exactly the same order as (1,2,...,340K) which may ease the model to over-fit the data. When random shuffle is on, each epoch starts with generating a random permutation of (1,2,...,340K) and feeds the data in the permuted order. Note that this does not affect validation and testing data, and no validation and testing data appears before 2018.

To avoid misunderstanding, we rephrased the paragraph into: “The time resolution of the ERA5 data is 1 hour; in the training subset (1979-2017), there were as many as 341,880 time points, the amount of training data in one epoch. To alleviate the risk of over-fitting, we randomly permuted the order of the training data at the start of each epoch. We trained four deep networks with lead times (the time difference between input and output) at 1 hour, 3 hours, 6 hours, and 24 hours, respectively. Each of the four deep networks was trained for 100 epochs, which takes approximately 16 days on a cluster of 192 NVIDIA Tesla-V100 GPUs.” Also, we enriched the description in the appendix (see “Optimization details”) using “All starting time points in the training subset (1979-2017) were randomly permuted in each epoch to alleviate over-fitting.”

In addition, to validate that Pangu-Weather produces stable forecasts throughout time, we added the testing results in 2020 and 2021 to the appendix. Note that the 2020 and 2021 data appears later than the 2019 data where we used sparsely for validation. This validates that Pangu-Weather has the ability to work on future testing data, i.e., Case (b) in Figure 4 of (Schultz et al., 2021).

(2) We agree that Pangu-Weather was trained and tested on reanalysis data and thus its skill of true forecast needs further investigation. We tried to bridge the gap and had some preliminary results by replacing part of the reanalysis input with IFS initial condition data in the testing phase (see below). We look forward to your advice on whether to add these results to the paper.

Here is how it works. The training phase remains unchanged, but in the testing phase, we replace the reanalysis data on 8 out of 13 pressure levels (200hPa, 250hPa, 300hPa, 500hPa, 700hPa, 850hPa, 925hPa, 1000hPa) and the surface level with the IFS initial condition data (we downloaded the data from TIGGE), and remain using the reanalysis data on other 5 levels (50hPa, 100hPa, 150hPa, 400hPa, 600hPa) because the IFS initial condition data is not available. The results show very small drop in forecast accuracy, indicating that our algorithm has the potential of training on reanalysis data and testing directly on IFS initial condition data.

During the revision, we added a paragraph to the end of the paper, discussing the limitations of our algorithm. It contains the following words: “Despite the promising forecast accuracy on reanalysis data, our algorithm has some limitations. First, throughout this paper, Pangu-Weather was trained and tested on reanalysis data, but real-world forecast systems work on observational data. There are differences between these data sources; thus, Pangu-Weather’s performance across applications needs further investigation. Second, some weather variables, such as precipitation, were not investigated in this paper. Omitting these factors may cause the current model to lack some abilities, e.g., making use of precipitation data for the accurate prediction of small-scale extreme weather events, such as tornado outbreaks. Third, AI-based methods produce smoother forecast results, increasing the risk of forecasting extreme weather events. We studied a special case, cyclone tracking, but there is much more work to do.

Fourth, temporal inconsistency can be introduced by using models with different lead times. This is a challenging topic worth further investigation.”

(3) We agree that the manuscript needs to down tone the claims. For detailed changes, please refer to our response to the **L.20** and **L.69** entries in **Q5**.

If necessary, we are happy to offer further explanations about these issues.

Q4: *The language needs polishing by a native speaker. For example, single case and plural are sometimes mixed in one sentence. While I corrected some language errors, I did not pay full attention to these in all parts of the manuscript.*

A4: Sorry for the language issues and thanks for the careful proofreading. We invited a native English speaker from SimpleTense (<https://www.simplentense.com/>), an academic editing service website, to polish the manuscript.

Q5: *Minor comments.*

A5: Thanks again for the detailed comments. We provide a point-by-point response below. Please note that the line numbers in the questions (entries) refer to the original version, while the line numbers in the answers, unless specified, refer to the revised version.

L.7: *“formulate the physical rules of atmospheric states” is a rather sloppy formulation. The PDEs rather describe the transition between atmospheric states. Also worthwhile to mention the discretisation of NWP models.*

Response: During the revision, we changed the description accordingly, including:

- **L.7** “Currently, the most accurate forecast system is the numerical weather prediction (NWP) method, which shows atmospheric states as discretized grids and numerically solves partial differential equations (PDEs) that describe the transition between those states.”
- **L.25** “Conventional NWP methods are primarily concerned with describing the transitions between discretized grids of atmospheric states using partial differential equations (PDEs) and then solving them with numerical simulations.”

L.20: *Please downtone the conclusion a little bit. There are still a few steps to be taken, before AI models are really better than NWP. In particular, as mentioned above, Pangu Weather needs to be run in a true forecast setting with input data it has never seen during training. Also, NWP models make direct use of observational data, while Pangu Weather requires a NWP model field to issue a forecast. Rather than “progress of AI surpassing NWP” it should say something like “progress on the path of establishing AI as a complement or surrogate of NWP” - the remark that it was thought to be far away should be moved to the Introduction and removed from the abstract, because it is still a bit away.*

Response: We agree. We down toned the wording throughout the manuscript. We summarize the changes below. Further comments are much welcomed.

- **L.11** “Here we introduce an AI-based method that, for the first time, surpasses NWP methods in terms of forecast accuracy” → “In this paper, we introduce an AI-based method for accurate,

medium-range global weather forecasting”.

- **L.15** “our program Pangu-Weather outperforms world’s best NWP system, operational IFS of the European Centre for Medium-range Weather Forecasts (ECMWF)” → “our program, Pangu-Weather, is the first to obtain stronger deterministic forecast results on reanalysis data in all tested variables, when compared with the world’s best NWP system, the operational integrated forecasting system (IFS) of the European Centre for Medium-Range Weather Forecasts (ECMWF)”.
- **L.20** “outperforms ECMWF-HRES in tracking tropical cyclones” → “When initialized with reanalysis data, the accuracy of tracking tropical cyclones is also higher than ECMWF-HRES”.
- **L.46** “surpasses existing NWP methods in terms of forecast accuracy of all tested weather variables” → “is the first to produce stronger deterministic forecast results than operational IFS on all tested weather variables against reanalysis data”.
- **L.77** “advances the progress of AI surpassing NWP” → “advances the progress on the path toward establishing AI as a complement or surrogate of NWP”. This sentence was moved from the end of abstract to the end of introduction.
- **L.223** “surpasses world’s best NWP system, operational IFS of ECMWF, in terms of both accuracy and speed” → “produces better deterministic forecast results on reanalysis data than the world’s best NWP system, operational IFS of ECMWF, while also being much faster”.
- **L.229** “Despite the unprecedented accuracy” → “Despite the promising forecast accuracy on reanalysis data”. In what follows, we discussed on the limitations of AI-based methods.

L.22: wording: “application” instead of “scenario”.

Response: It has been modified accordingly.

L.24: “high performance computers” instead of “the ... device”.

Response: We used “high-performance computing systems” as Referee #3 suggested.

L.28: imprecise wording “parameterisations”, not “parametric models”; it should be said that parameterisations are needed to capture subgrid-scale (or unresolved) processes. As I suggest to delete the final discussion on NWP versus AI-based models in the annex, you could re-use some of the information from there to state more clearly what empirical parameterisations are.

Response: We rephrased the sentence into: “Additionally, conventional NWP algorithms largely rely upon parameterization, which uses approximate functions to capture unresolved processes, where errors can be introduced by approximation.” Suggestions are much welcomed.

L.33: equating “input” with “observed data” gives a wrong impression that you were using observations as inputs, while in fact this was reanalysis data, hence, ultimately, a (NWP) model result.

Response: We have changed “input (observed data)” into “input (reanalysis weather data at a given point in time)” and “output (target data to be predicted)” into “output (reanalysis weather data at the target point in time)”. Another inaccurate statement later (L.80) was also revised from “training

deep neural networks that take observed weather data as input and produce future weather data as output” to “training deep neural networks to take reanalysis weather data at a given point in time as input, and then produce reanalysis weather data at a future point in time as output”.

L.47: this could be confusing to readers not familiar with the AI literature: if 3D “can better capture the relationship between pressure levels”, then one may wonder “compared to what baseline?”. Have earlier models not had more than one level? Or did they have no control over the coupling?

Response: By “3D models are better”, we meant to compare it against 2D models, such as what have been used in FourCastNet. To be clear, FourCastNet trained a 2D model to deal with all 20 variables from five pressure levels (see Table 1 in FourCastNet). After the first convolutional layer, the 5 pressure levels are indistinguishable from each other, losing useful information that helps weather forecast. We used 3D models which formulate the height information (in pressure levels) in an individual dimension. To make things clearer, we revised the original contents into: “Our experiments show that 3D models, by formulating height into an individual dimension, have the ability to capture the relationship between atmospheric states in different pressure levels and thus yield significant accuracy gains, compared to 2D models such as FourCastNet.”

L.60: I wouldn’t call 297 “much” lower than 333. Please delete “much”.

Response: It has been deleted. We also avoided the use of “much” in L.38, L.174, L.452, and L.458 (of the original version) to down tone the claims.

L.62: why provide the time in ms when other values are given in s or h? Replace with “1.4 s”.

Response: It has been modified accordingly. We also modified “280ms” into “0.28s” in L.444, and “1,400ms” into “1.4s” in L.445.

L.63: “sufficient details”: sufficient for what purpose. Clearly, it can be seen that PanguWeather forecasts are more blurred than ERA5 fields. It might be worth explaining the “bluriness” a bit more: the cyclone tracking images in the annex suggest that, at least for some fields, Pangu Weather results are not smoothed at all.

Response: We removed the sentence “Our visualization shows that Pangu-Weather produces sufficient details at a spatial resolution of $0.25^\circ \times 0.25^\circ$.” Instead, we modified the next sentence into: “Pangu-Weather not only produces strong quantitative results (e.g., RMSE and ACC), but also preserves sufficient details for investigating certain extreme weather events. To demonstrate this capability, we studied an important application of tropical cyclone tracking.”

L.65: “cyclone tracking” is an application, not a scenario.

Response: It has been modified accordingly.

L.66: Be careful with the claim that PanguWeather is the first DL model that can accomplish cyclone tracking. See, for example, <https://www.frontiersin.org/articles/10.3389/fdata.2020.00001/full> and <https://gmd.copernicus.org/articles/14/107/2021/gmd-14-107-2021.html>.

Response: Thanks for the reminder. We removed the sentence “This is the first time that AI-based methods are used for this scenario”.

L.69: please use less “dramatic” language. It would be more interesting to readers to get to know in which way the ECMWF tracking failed. Was it the typhoon path, its intensity, the timing?

Response: Agree. We avoided using the “dramatic” language in the revision. Specifically,

- **L.75** We changed “... that even ECMWF-HRES, world’s best tracking system, failed dramatically” into “... that remain a challenge for the world’s best tracking systems, such as ECMWF-HRES”.
- **L.192** We changed “before which the path is dramatically incorrect” into “before which it predicts that Yutu will make a big turn to the northeast”.
- **L.502** We changed “while the forecast of ECMWF-HRES is dramatically incorrect (Yutu makes a big turn and heads to the northeast)” into “while ECMWF-HRES incorrectly predicts that Yutu will make a big turn to the northeast in the early stage”.

L.72: it is technically not fully correct to refer to the ERA5 input data as “observations”. Substantial processing including error corrections went into observations and the IFS model acts as a sort of filter (i.e. data assimilation). Thus, while strongly bound by observations, the input data you have used are not directly observations.

Response: We made it clearer. The sentence now reads: “The methodology involves training deep neural networks to take reanalysis weather data at a given point in time as input, and then produce reanalysis weather data at a future point in time as output.”

L.74-77: This description looks like you applied random sampling of individual hourly values. Schultz et al. (2021), your reference #4, discussed the problems with this method. Basically, random sampling makes your model evaluation less meaningful due to auto correlation in the data.

Response: This might be a misunderstanding. Please refer to **Q3** (1) for detailed explanations. We revised the contents to make the details clearer.

L.88: What exactly do you mean with “Earth-specific positional bias”? Please add a reference to the extra material, where a description is given.

Response: We inserted “(a new mechanism of encoding the position of each unit; detailed in the Methods section)” after the phrase “Earth-specific positional bias”.

L.81-91 and Figure 1: The network architecture is interesting and overall well described. However, it may be useful to add another panel to Figure 1, or modify panel b, to document more clearly, how the 4 networks interact, if at all. From the panel b it appears that they run completely independently, but this cannot be, because the 54th hour in your example, of course depends on the first two iterations of the 24-hour model, i.e. there must be some sequential coupling. You do mention this implicitly in l. 103, but please make it more explicit.

Response: The four deep networks were individually trained and they interacted only in the testing phase. Let the input reanalysis data be $\mathbf{A} = \mathbf{A}_0$ where the subscript indicates the lead time. Given a target lead time, we use a greedy algorithm to iteratively find the trained model with longest lead time that can be used. This can maximally reduce the number of iterations and alleviate accumulative forecast errors. For example, if the desired lead time is 56 hours, we feed \mathbf{A}_0 to the

24h model and obtain \hat{A}_{24} (the hat indicates forecast results), then feed \hat{A}_{24} to the 24h model again to obtain \hat{A}_{48} . Then, the 6h model is called 1 time to obtain \hat{A}_{54} , and the 1h model is called 2 times to obtain \hat{A}_{55} and \hat{A}_{56} , with \hat{A}_{56} being the final output. We modified Panel (b) of Figure 1 to better illustrate the described procedure.

L.140-147: This is a somewhat awkward discussion about the model smoothness. It is a general property of any regression algorithm (and a large transformer network is nothing else) to converge to average values. In contrast, a single IFS forecast calculates a single estimated value at each grid cell and for each time step by solving a system of PDEs with initial conditions. Due to the chaotic nature of weather and the inevitably imprecise knowledge of the initial conditions and subgrid scale processes, weather forecasters have long known that there are statistical uncertainties associated with each forecast, and they have thus introduced the concept of ensemble forecasting. Overly smooth fields can never be seen as advantage.

Response: We totally agree with your comments. To be clear, we did not mean that smooth fields can be seen as advantage (sorry for misleading), so we mentioned the ability of Pangu-Weather to “preserve small-scale structures” and further validated it in tracking tropical cyclones. Thanks for your professional comments that help us to make these words clearer.

We revised the discussion into: “It is a general property of any regression algorithm (including deep neural networks) to converge average values. In contrast, the operational IFS forecast is less smooth, because it calculates a single estimated value at each grid cell by solving a system of PDEs with initial conditions, while the chaotic nature of weather and the inevitably imprecise knowledge of the initial conditions and sub-grid scale processes can cause statistical uncertainties in each forecast.” We removed the statement that “Pangu-Weather preserves sufficient details” and leave the discussion to the next part where we used Pangu-Weather to track tropical cyclones.

L.155: This statement doesn't make sense: extreme events are characterized as being at the tails (usually upper tail) of a variable distribution. Their relevance stems from the fact that they are extreme, and “accuracy” can only refer to getting the magnitude right. It can make an enormous difference for people whether a rain event generates 80 or 120 mm of rain. Of course, there are other relevant evaluation criteria for extreme events, such as the correct timing and advection path. These must be discussed separately.

Response: We agree that RQE may not fully reflect the ability of extreme weather forecast. We removed the entire paragraph from the main article and changed the section title from “extreme weather events” to “tracking tropical cyclones” accordingly. The paragraph of RQE is preserved in the appendix. We also provided an improved visualization of the quantiles (see Extended Data Figure 8) as suggested by Referee #2.

L.158: “specific treatment of extreme weather data” is too unspecific. This remains one of the most difficult challenges for applying AI trustfully to weather data. Unless you document some ideas, this statement should be better formulated as a remaining challenge rather than an expectation.

Response: While we do have some preliminary ideas (see below), we agree that it is necessary to notice the readers that “AI-based methods produce smoother forecast results, increasing the risk of forecasting extreme weather events. We studied a special case, cyclone tracking, but there is much more work to do.” We added it to the discussion of limitations.

Our idea is based on the fact that large-scale AI models have an ability known as “few-shot learning”, i.e., adjusting to specific data distributions based on small data. We hope that, with the help of meteorologists, we can collect some datasets for specific extreme weathers (e.g. tropical cyclones) which, albeit being smaller than the full reanalysis data, allow us to fine-tune AI models on these data to improve the ability of extreme weather forecast.

L.160-173: The typhoon tracking results are impressive. However, as stated in my major comment, the comparison is somewhat unfair, because the ECMWF HRES model had no knowledge about the future, whereas Pangu Weather was trained on data with more frequent input from observations, and the tracking results have not been generated with completely independent input data, i.e. data that has never been seen during training or fine-tuning.

Response: First, please kindly refer to **Q1** to avoid misunderstandings. Pangu-Weather was trained on 1979-2017 data and all the tested cyclones appear in or after 2018. In other words, Pangu-Weather was not “interpolating between training data” because weather data in or after 2018 was not seen during training (the validation data in 2019 was used only one time to check if the testing results look fine).

Indeed, Pangu-Weather was tested on reanalysis data, so we added a paragraph to the end of this section saying: “Despite the promising tracking results, we point out to the reader that the direct comparison between Pangu-Weather and ECMWF-HRES is somewhat unfair, because ECMWF-HRES used IFS initial condition data as its input, while Pangu-Weather used reanalysis data.” This limitation was also added to the discussion part in a wider scope (covering both deterministic forecast and tracking tropical cyclones).

L.172 ff: Please add more specific information to the Perlin noise perturbation that was used to generate the ensemble forecasts. For example: what was the maximum length of the noise vector? How were vector lengths distributed?

Response: In the main article, we simplified the description saying “We then generated 99 random perturbations (detailed in the Methods section) and added them to the unperturbed initial state.” Meanwhile, in the appendix, we offered details saying: “Each perturbation generated for ensemble weather forecast contains 3 octaves of Perlin noise, with the scales being 0.2, 0.1 and 0.05, and the number of periods to generate along each axis (the longitude or the latitude) being 12, 24 and 48, respectively. We used the implementation provided in a GitHub repository, <https://github.com/pvigier/perlin-numpy>, and modified the code for acceleration.” We also inserted a section in the pseudocode to better clarify the details.

L.205: As stated above, the claim that “Pangu-Weather, for the first time, surpasses world’s best NWP system” is over stated.

Response: We changed the sentence from “surpasses world’s best NWP system, operational IFS of ECMWF, in terms of both accuracy and speed” → “produces better deterministic forecast results on reanalysis data than world’s best NWP system, operational IFS of ECMWF, meanwhile being much faster”. We are more than happy to receive further comments if necessary.

L.214: What do you mean by “observation factors”? Are these additional variables on more vertical levels?

Response: Both can help. Currently, we followed WeatherBench to use 13 out of 37 pressure levels and chose to report the weather variables that were published in TIGGE. Adding more variables requires larger GPU memory which we will try in the future. To be clear, we modified “incorporating more observation factors” into “incorporating more vertical levels and/or atmospheric variables”.

L.289: please insert a reference to the “extended data table 1” in the annex.

Response: Done. We inserted a sentence to the “Data preparation details” subsection saying: “For a complete list of studied variables and the corresponding abbreviations, please refer to Extended Data Table 1.”

L.295: The evaluation remains a bit shallow if compared to the established standards in numerical weather modeling. See for example <https://sites.ecmwf.int/ifs/scorecards/scorecards-47r3HRES.html> for a regional breakdown of results. At a minimum, I suggest that the metrics are evaluated also for Northern and Southern hemisphere separately. It would also be highly beneficial to get a separate metric for the tropics, because this is the region where classical numerical models have the largest uncertainty.

Response: We followed the suggestion to add three new figures (Extended Data Figures 1-3), showing the quantitative comparison in the Northern Hemisphere, the Southern Hemisphere, and the tropics against operational IFS (we did not compare against FourCastNet because it did not report these breakdown results). The texts were modified accordingly:

- **L.138** We added a sentence: “For quantitative studies in the Northern Hemisphere, the Southern Hemisphere, and the tropics, please refer to the Extended Data Figures 1-3.”
- **L.338** We added a sentence: “The RMSE and ACC metrics can also be broken down into specific regions, e.g., in the Northern Hemisphere, the Southern Hemisphere, and the tropics. Please refer to Figure 2 and Extended Data Figures 1-3 for the overall and breakdown results in 2018.”

L.305 ff: Here, another citation of Hersbach et al., (2020) would be appropriate.

Response: It has been added.

L.355: Please include a reference to Swin transformer here, because the window shift is a key feature of Swin T.

Response: It has been added.

L.356: “Along the longitude ...” this sentence is incomprehensible.

Response: This is a small trick in implementation. For example, if there are 1440 units along the longitude and they are partitioned into 4 windows so that each window occupies 360 units. If we shift the partition by half a window, there will be 3 entire windows in the middle and 2 half windows in both ends. But, since the 2 half windows are actually adjacent to each other, they are directly merged into one window. Note that we cannot do this along the latitude.

We revised the contents into: “There is a special treatment here. Note that the Earth’s lines of latitude are circular and thus both ends along the longitude are close to each other in Earth’s surface.

In the shifted-window mechanism, if half windows appeared at both leftmost and rightmost positions, they would be directly merged into one window. The merge operation was not performed along the latitude dimension because the two ends (the North Pole and the South Pole) are not adjacent to each other.”

L.367: delete “modify”.

Response: It has been fixed.

L.371: the “neurons” are not partitioned; please use the term “data volume” or similar instead.

Response: We meant that “the set of neurons are partitioned”. We agree that “data volume” is more accurate and replaced the original content accordingly.

L.376: confusing: the window coordinate should be only (mpl, mlat) if mlon is not used. You could also write “2-dimensional window coordinate”.

Response: Fixed. We revised it into “we used the first two indices of the window coordinate, (m_{pl}, m_{lat}) , to locate the corresponding bias sub-matrix ”.

L.380: Even though it is understandable that you cannot fully train a large number of network configurations, it would nevertheless be illustrative to show loss curves or accuracy metrics of at least some variants over a couple of training epochs and discuss the observed differences.

Response: Done. We added a sentence to the end of the next paragraph (“Optimization details”) saying: “We plotted the accuracy of some tested variables with respect to different lead times (1h, 3h, 6h, 24h) in Extended Data Figure 7.”

L.390: Is there any evidence to support your believes that 4D tensors would lead to better results? Have you run tests with smaller configurations? Is there a paper in the literature that proves this for other types of data?

Response: Yes. 4D deep networks were investigated in the computer vision community, showing better performance than 3D counterparts. We provide two examples below. They have been cited in the revised manuscript.

[a] Choy, C., Gwak, J. Y., & Savarese, S. 4d spatio-temporal convnets: Minkowski convolutional neural networks. In *Proc. IEEE/CVF Conf. Comput. Vis. Pattern Recognit.* 3075-3084 (2019).

[b] Zhang S., Guo, S., Huang, W., Scott M. R., & Wang, L. V4d: 4d convolutional neural networks for video-level representation learning. Preprint at <https://arxiv.org/abs/2002.07442> (2020).

L.401: I suggest to delete “while still being one of the fastest systems...”. It is sufficient to say that Pangu Weather is about 10.000 faster than IFS and of comparable speed as Fourcastnet.

Response: The sentence has been deleted.

L.406: “follow” instead of “following”.

Response: It has been fixed.

L.404 ff: The RQE does not evaluate if extreme values occur at the right location and time, but only looks at the tails of the value distribution. While the typhoon tracking results indicate that Pangu

Weather can also capture individual extreme events, it would be good to show this more explicitly.

Response: We agree that RQE has limitations. We computed it to compare against FourCastNet. During the revision, we weakened the display of RQE by removing it from the main article (the main article only studies cyclone tracking now) and we added one sentence after the description of RQE in the appendix: “Please note that RQE and the individual quantile values have limitations: they do not evaluate if extreme values occur at the right location and time, but only looks at the value distribution. The ability of Pangu-Weather to capture individual extreme events was further validated with the experiments of tracking tropical cyclones.”

L.432: This discussion of cyclone tracking results is unfair, because ECMWF-HRES had no information about the future when it had to issue the forecasts, whereas Pangu Weather was trained on ERA5 data with daily updated analyses. Furthermore, as no data was held back for the evaluation, Pangu Weather “only” had to interpolate between data it has seen during training. While the results are still very good, this is not a final verdict of the true forecasting quality of Pangu Weather. For this, you should evaluate cyclones in a year that was not used for training, e.g. 2022.

Response: Please refer to our response to the entry of **L.160-173**. We emphasized the nature of unfair comparison again in the appendix: “Again, we emphasize that the comparison against ECMWF-HRES is somewhat unfair, because ECMWF-HRES used IFS initial condition data, while Pangu-Weather used reanalysis data.”

L.467: what do you mean by “good practice”?

Response: We meant “good results”. In the revised version, “the good practice of deterministic forecast” was rewritten as “the accurate deterministic forecast results on reanalysis data”.

L.468 ff: This summary of previous work is somewhat awkward. The issues with NWP models have been discussed in other papers, and the benefits or approaches of AI-based methods are not explained well enough to convince readers. What is required to be known about the AI approach and specifically Pangu Weather is (and should be) contained in the main text and the detailed network and data descriptions above. I don’t think, this section is necessary and the manuscript would be better without it.

Response: Respectfully, we think that having this part is beneficial for the readers to compare the NWP methods with AI-based methods. So, we discussed this issue with the editor and he suggested that we maintain this section but make it shorter. We followed the advice and reduced the contents by about 50%.

*L.520: “code base *of* ...”*

Response: It has been fixed.

L.597: (extended data table 1): please add the levels from which data were extracted in the table caption and the total number of resulting variables so that all relevant information is accessible at one place.

Response: Done. Now, the caption says: “The correspondence of upper-air and surface variable names and their abbreviations. Throughout the paper, we extracted the upper-air variables from 13 out of 37 pressure levels (50hPa, 100hPa, 150hPa, 200hPa, 250hPa, 300hPa, 400hPa, 500hPa,

600hPa, 700hPa, 850hPa, 925hPa, 1000hPa) plus the surface variables. Therefore, a total of 69 variables were studied in the experiments.”

Q6: *Code and data availability: I could trace back the data sources and ran the code provided at the link that was made available to the reviewers. This worked well, and the code is well structured. The only comment there would be to collect the library requirements in a requirements.txt file instead of listing them in the README. Furthermore, the authors should of course release their code publicly when the paper is accepted and before publication.*

A6: Received. As said in **Q1** to the editor, we have created a GitHub repository where we will release the pseudocode and trained models after the paper is accepted. An individual file, “requirements.txt”, will be provided to ease the preparation of the dependencies.

Response to Referee #2

Q1: In this manuscript the authors tackle medium range weather forecasting using machine learning. They develop a 3D transformer model with the ability to learn location specific terms and train on the ERA5 archive. The model shows strong results for both headline scores and tropical cyclones, indicating that a useful tool has been learnt. The task itself is not novel, but the model has novelty. The results themselves are a significant step beyond previous results. This work will, in my opinion, make people reevaluate what forecasting models might look like in the future.

A1: Thanks for the summary on our manuscript.

Q2: On data and methodology, the data used for training and evaluation is appropriate. There are two issues with the analysis methodology. The section on ensemble forecasting completely ignores a key point, that ensembles should be assessed through probabilistic scores, not deterministic ones. Figure 5 should be replaced. My suggestion is that the RMSE plot be replaced by a CRPS plot (which is defined in equation 4 here <https://arxiv.org/pdf/2205.00865.pdf>), this could still compare the deterministic and ensemble models, as the CRPS for a deterministic forecast reduces to the MAE. This would be a valuable addition to the community as future projects can then compare against Pangu's results. I would additionally suggest the authors plot the spread/error ratio (defined in the same paper as equations 1 & 3) as a function of lead time. This shows the reliability of the forecast. Secondly, ROE as a single number lacks interpretability and does not add value. This data would be best shown by showing the humidity value distribution rather than aggregating to a single number, for example a plot similar to the left panels in FourCastNet's figure 9. This should be shown for a lead time such as 3 or 5 days, perhaps both.

A2: Good suggestion! We followed the suggestions to add CRPS values to the RMSE plots in Figure 5 and use the spread-error ratio plots to replace the ACC plots. The citation was added accordingly. Also, we followed the left panels in FourCastNet's Figure 9 to add the quantile plots for all tested variables as Extended Data Figure 7 and modified the contents in the appendix accordingly.

Q3: The broad conclusions are appropriate but they ignore several important points which I think should be added. While, I agree with the authors that there is room for improvement on the AI side. I think it is worth noting that there is also room for improvement on the conventional modelling side. NWP models develop biases which are predictable and therefore can be compensated for with post-processing of the NWP output. In the search for the optimal forecasting model it is still possible that a combination of NWP and ML-based post-processing produces the optimal product. I would like to see this noted in the conclusions. Perhaps most importantly, ERA5, whilst of high quality is not a universal truth. A true test of any model would be against quality-controlled observations. These are as easily accessible as ERA5, so suggesting a comparison is not within the scope of the paper. However this qualification should be stated at key moments in the paper. For example the abstract should read "...surpasses NWP methods in terms of forecast accuracy when measured against reanalysis." In the discussion this comparison against observations should be highlighted as vital future work to further establish the skill of Pangu-Weather.

A3: We agree with these comments and made the following changes accordingly.

- We appended the following sentence to the final paragraph of the discussions: "On the NWP side, post-processing methods can be developed to alleviate the predictable biases of NWP models. We

expect that AI-based and NWP methods will be combined in the future to bring about stronger performance.”

- We followed the suggestions of Referee #1 to down tone the wording and add constraints to our statements. For example, the sentence in abstract was rephrased into “obtain stronger deterministic forecast results on reanalysis data in all tested variables, when compared with the world’s best NWP system, the operational integrated forecasting system (IFS) of the European Centre for Medium-Range Weather Forecasts (ECMWF)”. For other changes, please refer to the response to Referee #1 (the **L.20** entry in **Q5**).
- We added a conclusive remark to the end of main paper on the gap between reanalysis data and observational data: “First, throughout this paper, Pangu-Weather was trained and tested on reanalysis data, but real-world forecast systems work on observational data. There are differences between these data sources; thus, Pangu-Weather’s performance across applications needs further investigation.”

***Q4:** The references are in general correct, although on L149 I believe this citation is incorrect and should be for FourCastNet.*

A4: This citation is correct. RQE was introduced in [Fildier et al., 2021], and FourCastNet set $D = 50$ quantiles. In the revision, follow the suggestion of Referee #1 to remove the RQE paragraph from the main article. In the appendix, we clearly stated that we followed [Fildier et al., 2021] to compute RQE and followed FourCastNet to set the percentile values.

***Q5:** In the authors’ manuscript on a public archive system they included a breakdown of TC track errors by region and intensity. I found this figure most interesting and would suggest it is added to the appendix of this work.*

A5: Received. It has been added as Extended Data Figures 10 and 11 in the revised version. We added a sentence to the main article: “The breakdowns of tracking errors with respect to regions and intensities are provided in Extended Data Figures 10 and 11.” Also, we added a new paragraph in the appendix: “We extended Figure 4c by plotting the mean direct position errors with respect to different basins or different intensities in Extended Data Figures 10 and 11. In each subset, Pangu-Weather reports lower errors and the advantage becomes more significant with a greater lead time, aligning with the conclusions we drew from the entire dataset.”

***Q6:** “ECMWF-HRES, world’s best tracking system, failed dramatically” -- this is an inaccurate representation of these events. TC tracks for both of these events were well-predicted, however not with as early of a warning as Pangu demonstrates. Both in the text and caption it should be noted that Pangu identifies the correct track earlier, rather than suggesting that the events were forecast busts for the IFS.*

A6: We agree. We changed “that even ECMWF, world’s best tracking system, failed dramatically” into “that remain a challenge for the world’s best tracking systems, such as ECMWF-HRES”. In the caption of Figure 4, the original version has made it clear that IFS obtains the correct path 2 days later than Pangu-Weather. In the case studies of cyclones in the appendix, we have also stated that “The advantage of Pangu-Weather mainly lies in tracking cyclone paths in early stages.”

Q7: Temporal consistency is an important feature of weather forecasts, for example rainfall over many hours will contribute to flooding (understanding that rainfall is not an output of PanguWeather). How does Pangu weather, by using multiple networks, ensure temporal consistency? For example the predictions for 23 and 24 hours are the product of different models/steps. I would label this an outstanding challenge in your conclusions.

A7: Good question! Indeed, temporal consistency is an important topic yet remains uncovered in this work. We added a sentence to the limitation paragraph, saying: “Fourth, temporal inconsistency can be introduced by using models with different lead times. This is a challenging topic worth further investigation.”

Q8: L124/5 It would be useful to have a table showing the “forecast gain time” of Pangu vs the baseline methods across the various variables.

A8: During the revision, we provided the suggested contents as Extended Data Table 2 and added a reference in the main article accordingly.

Q9: “This is a typical property of deep neural networks in learning from large-scale datasets” -- it should be noted that smooth predictions for some fields, particularly those associated with moist processes, can degrade the impact of forecasts, for example less extreme rainfall events than conventional models.

A9: We agree. To be clear, we did not mean that smooth predictions are good (sorry for misleading). Following the suggestion of Referee #1, we rephrased this part into: “It is a general property of any regression algorithm (including deep neural networks) to converge average values. In contrast, the operational IFS forecast is less smooth, because it calculates a single estimated value at each grid cell by solving a system of PDEs with initial conditions, while the chaotic nature of weather and the inevitably imprecise knowledge of the initial conditions and sub-grid scale processes can cause statistical uncertainties in each forecast.”

Q10: “Still, Pangu-Weather produces high-resolution forecasts that are very close to the ground-truth and preserve most of small-scale structures of surface variables.” -- how are you measuring or assessing this?

A10: Currently, an effective way to measure the ability of Pangu-Weather to preserve “small-scale structures” for surface variables is to investigate its forecast ability in some extreme weather events, e.g., tracking tropical cyclones. Since the subsequent paragraph shows the ability of cyclone tracking, we removed the statement from the manuscript to avoid ambiguity and/or misunderstanding.

Q11: “(1) incorporating more observation factors” I don’t understand this point. Please could you reformulate?

A11: Currently, we followed WeatherBench to use 13 out of 37 pressure levels and chose to report the weather variables that were published in TIGGE. Adding more variables requires larger GPU memory which we will try in the future. To be clear, we modified “incorporating more observation factors” into “incorporating more vertical levels and/or atmospheric variables”.

Q12: Minor comments: L9 potentials -> potential.

A12: Thanks. This bug has been fixed. We also invited a native speaker to polish the manuscript.

Response to Referee #3

Q1: Summary, Novelty/Originality, Data & methodology, References, Clarity and context.

A1: Thanks for the summary and positive comments on our manuscript. In what follows, we do not copy the reviews that do not require explicit answers to simplify the response letter.

Q2: (#1) Your weather forecast model does not predict precipitation! Precipitation is a crucial part of weather forecasts. It seems odd to evaluate a global weather forecasting system that does not even try to provide precipitation forecasts.

A2: Yes. When we were planning for the project in August 2021, we noticed that the ERA5 dataset said that the precipitation data may be more inaccurate than other factors. So, we decided not to test on this factor and did not download it from the official website of ERA5. After we read the precipitation forecast results by FourCastNet, we tried to download this part of data, but the link speed became very slow and we did not get it prepared before the internal project deadline.

We added a sentence to the section of discussions, saying “Second, some weather variables, such as precipitation, were not investigated in this paper. Omitting these factors may cause the current model to lack some abilities, e.g., making use of precipitation data for the accurate prediction of small-scale extreme weather events, such as tornado outbreaks [citing the mentioned reports].”

Q3: (#2) Not predicting precipitation also implies that the forecasts of your other variables cannot utilize precipitation estimates at prior time steps. I would expect the lack of precipitation data to cause problems for the accurate prediction of small-scale extreme weather events (such as tornado outbreaks, see <https://doi.org/10.1175/MWR-D-21-0013.1>, <https://doi.org/10.1175/MWR-D-21-0014.1>). That might be why both FourCastNet and Pangu-Weather focus on case studies for large-scale extreme events, such as hurricanes. I don't expect you to fix this problem, but at least discuss this limitation, please.

A3: Please see the response to **Q2**.

Q4: (#3) You only provide results using ERA5 data, which is not available in real-time. How well does your model work when you feed in real-time (e.g. GFS) data instead?

A4: Referee #1 raised the same question, where he/she suggested to try on IFS initial condition data. We did some preliminary tests on IFS initial condition data (GFS data is not available for us), and we copy the response here.

We agree that there is a gap between Pangu-Weather (working on reanalysis data) and true forecast (working on real-time data). We tried to bridge the gap and had some preliminary results, but we are not sure whether we should put them into the manuscript. Here are our tests. The training phase remains unchanged, but in the testing phase, we replace the reanalysis data on 8 out of 13 pressure levels (200hPa, 250hPa, 300hPa, 500hPa, 700hPa, 850hPa, 925hPa, 1000hPa) and the surface level with the IFS initial condition data (we downloaded the data from TIGGE), and remain using the reanalysis data on other 5 levels (50hPa, 100hPa, 150hPa, 400hPa, 600hPa) because the IFS initial condition data is not available. The results show very small drop in forecast accuracy, indicating that our algorithm has the potential of training on reanalysis data and testing directly on IFS initial condition data.

During the revision, we added a paragraph to the section of discussions, discussing the limitations of our algorithm. The first point is on the lack of investigation on real-time data: “Despite the promising forecast accuracy on reanalysis data, our algorithm has some limitations. First, throughout this paper, Pangu-Weather was trained and tested on reanalysis data, but real-world forecast systems work on observational data. There are differences between these data sources; thus, Pangu-Weather’s performance across applications needs further investigation.”

Q5: (#4) Reproducibility / advancing science: releasing pseudo-code and fully-trained models. Will you accept the challenge to make your model available in a similar form?

A5: Yes. As we promised to the editor (we copy the response here), we are willing to share the pseudocode and trained models in a GitHub repository assigned with a DOI. Please check the link and information below. The repository is temporarily empty, because we need a permission from our company to send out models from our internal network system. The paperwork can take a few weeks or so, and we will surely get it ready before the paper is published. We added the information to the Code availability section.

GitHub repository: <https://github.com/198808xc/Pangu-Weather>

DOI: 10.5281/zenodo.7654468

Q6: It would be fun to include a paragraph on why this model is named “Pangu”. I assume it’s named after the Chinese mythology figure “Pangu” (<https://en.wikipedia.org/wiki/Pangu>)? Would be great to provide a bit of cultural context/education to the audience.

A6: Good idea. Yes, the name of Pangu comes from the Chinese mythology figure. We added a short paragraph to the end of the appendix and inserted a reference into the place that Pangu first appears in the introduction part.

The short paragraph says: “Pangu is a primordial being and creation figure in Chinese mythology who separated heaven and earth and became geographic features such as mountains and rivers (see the Wikipedia page: <https://en.wikipedia.org/wiki/Pangu>). Pangu is also a series of pre-trained AI models developed by Huawei Cloud that covers computer vision, natural language processing, multimodal understanding, scientific computing (including weather forecasting), etc.”

Q7: The English in the main document needs some polishing – the meaning is generally clear but the English is clumsy. Example: high-performance computational device -> high-performance computing system.

A7: We invited a native English speaker from an academic editing service website, SimpleTense (<https://www.simpletense.com>), to polish the manuscript. Regarding the mentioned example, it has been fixed accordingly.

Reviewer Reports on the First Revision:

Referees' comments:

Referee #1 (Remarks to the Author):

Thanks to the authors for carefully considering all reviewer comments and providing an updated, improved version of the manuscript that avoids misunderstandings and includes even more great results. With this, I am convinced that the paper makes an important contribution to the field and that it is scientifically and technically sound.

A number of minor comments are listed below. In DL lingo these can be considered "fine-tuning".

l.85 OK. However I would suggest to write even more clearly "..., we randomly permuted the order of *samples from* the training data..."

l. 100 please remove the "X" - the word "factor" suffices

Figure 1: nice improvement, much clearer now. Another minor suggestion: "24 hours model" etc sounds a tiny bit clumsy. Perhaps introduce labels such as FM1, FM3, FM6, and FM24 for "forecast model with 1, 3, 6, and 24 hours lead time, respectively"?

l. 114 Please include a reference to Figure 1b here.

Figure 2: please increase line thickness in the legend. I had to zoom in considerably to distinguish blue from black. The panels themselves are OK.

l. 148 suggest to rephrase "... technical design, resulting especially from the 3D ..."

l. 162 "converge *on* average values"

Figure 5: figure quality is low. And please increase line thickness in the legend.

l. 236 you probably wanted to say "results, increasing the risk of forecasting extreme weather events *inaccurately*" or ", increasing the risk of underestimating the magnitudes of extreme weather events."

l. 242 It might be good to add the references you provided in the rebuttal to the mentioning of 4D models.

l. 247 "bring about *even* stronger performane."

l. 329 ff. Please add short definitions of the CRPS and the other new scores you have introduced.

I. 338 I suggest to rephrase "The RMSE and ACC metrics can also be evaluated for specific regions..."

I. 398 The special treatment could be described a bit more succinctly. For example: "... by half window size. Since coordinates in longitude direction are periodic, the half windows at the left and right edges are merged into one full window."

I. 435 two "limits". Suggest: "the limited available computational budget prevented us from exploring this method."

I. 467 "only look" (delete "s")

I. 535 Parametrizations were not introduced primarily to "accelerate" but rather to achieve "closure", i.e. solving the differential equations on scales that are smaller than the model resolution. Side note: there are interesting implications with respect to parametrizations when model resolution is increased substantially. For example, parametrizations of convection usually assume that all transport of air occurs within one vertical column of grid boxes. However, once you go below ~5 km, you must take into account horizontal exchange between neighbouring boxes - and this means that convection directly impacts on the (spatially resolved) dynamics of the model. This is only one of the challenges for further advancing traditional NWP models.

I. 541 suggestion: "offer a complementary path"

I. 545 "such as short-term precipitation..." - the point here is that on time scales up to a few hours, the traditional models are too much influenced by the initial conditions. Classical "nowcasting" methods have used traditional imaging algorithms (flow field extrapolation etc.), and these could be readily replaced by DL models.

I. 549 I am now happy with the shorter discussion about NWP versus DL. However, I would suggest to add another sentence at the end referring to the current state of DL with very large networks achieving an abstract representation of the training data, and that this goes far beyond earlier attempts based on U-Nets or GANs, etc.

I. 636 replace "studied in the experiments" with "used as inputs"

I. 658 Please include a definition of "tropics" in the figure caption. Did you use 15 or 20 degrees as latitude cutoff?

Out of curiosity: some panels, especially T2m show an interesting zigzag pattern. Do you have an explanation for this?

Extended data figures 4 and 5: to ease comparison, please add the 2018 results on these plots as well. (perhaps as dashed lines)

And finally a question: you mentioned that you ran Panguweather based on IFS initial states and would seek the reviewers' opinion on whether to include these results. However, I did not find them in the rebuttal letter nor the manuscript. In any case, I would suggest to leave the paper at its

present state and only, if you so wish, include a sentence somewhere to say "Preliminary results with using IFS initial conditions instead of ERA5 data as model inputs suggest that Panguweather will lose some forecasting capability, but not deteriorate dramatically."

Referee #2 (Remarks to the Author):

I thank the authors for their replies and changes, I think the manuscript is significantly improved. Below I outline a few minor changes I think will improve the paper. After adopting these changes I would be happy to see the manuscript published.

Figure 5: Please could the spread-skill plot include a dotted/dashed horizontal line indicating a ratio of 1 to highlight the ideal behaviour, and mention this in the caption.

Extended figures 4 & 5 are very hard to interpret. I would suggest having one figure, showing years 2018, 2020 & 2021 in different colours to allow comparison. It should be noted on the plot that predictability varies from year to year which is not compensated for in this plot. If you have IFS data for all of these years then instead a better plot would be to show the performance relative to the IFS for each of the years, by comparing to the IFS baseline this would compensate for the predictability variation. e.g. plotting $(RMSE_{pangu} - RMSE_{ifs})/RMSE_{ifs}$ for each of the years separately. Even this has some caveats, as IFS has improved over this period, it would at least give context to the Pangu scores.

Extended figure 8 appears to be very low resolution. I struggled to read it. I very much welcome its inclusion though.

Extended figure 10 has no label.

L88 is this 16 days and 192 GPUs per model?

I cannot find several interesting details on your training. Specifically the loss used, the normalisation of the training data, and any scaling of the variables or pressure levels within the loss (or are all normalised variables given equal weighting?). I see no discussion of the learning rate used, or any details on possible adaptive learning rate. Please specify this. Please include these.

L438 DropPath, please could the authors provide more detail about the specific implementation and settings that they used.

Referee #3 (Remarks to the Author):

I would like to cordially thank the authors for responding in this new revision to my recommendation to make the models publicly available to the community - in addition to providing the original pseudo code which is also very helpful for a deeper understanding of the model.

Our group picked one trained model, the 24h forecast model, and I can confirm that it is very easy to download and run it. It just took us one afternoon to get this to work, and it executed quickly on even a desktop computer.

This means that anyone in the meteorological community can now run and test these models to their heart's desire. What a great opportunity for the community to explore how well the model predicts specific meteorological phenomena. Now THAT's going to help with progress in the field.

Kudos to the author for making this possible. Thank you!

Author Rebuttals to First Revision: Response to Referee #1

Q1: Thanks to the authors for carefully considering all reviewer comments and providing an updated, improved version of the manuscript that avoids misunderstandings and includes even more great results. With this, I am convinced that the paper makes an important contribution to the field and that it is scientifically and technically sound.

A1: Thanks for the positive comment and all your help so that we could improve the manuscript.

Q2: A number of minor comments are listed below. In DL lingo these can be considered “fine-tuning”.

A2: Thanks again for the detailed comments. We provide a point-by-point response below. Please note that the line numbers in the questions (entries) refer to the original version, while the line numbers in the answers, unless specified, refer to the revised version.

*L.85: OK. However I would suggest to write even more clearly “..., we randomly permuted the order of *samples from* the training data...”*

Response: Done.

L.100: Please remove the “X” - the word “factor” suffices

Response: Fixed.

Figure 1: nice improvement, much clearer now. Another minor suggestion: “24 hours model” etc sounds a tiny bit clumsy. Perhaps introduce labels such as FM1, FM3, FM6, and FM24 for “forecast model with 1, 3, 6, and 24 hours lead time, respectively”?

Response: Done. We replaced the text labels in Figure 1b into “FM1”, “FM3”, “FM6”, and “FM24”, and updated the caption of Figure 1 accordingly: “We use ‘FM1’, ‘FM3’, ‘FM6’ and ‘FM24’ to indicate the forecast models with lead times being 1 hour, 3 hours, 6 hours, or 24 hours, respectively.”

L.114: Please include a reference to Figure 1b here.

Response: Done.

Figure 2: Please increase line thickness in the legend. I had to zoom in considerably to distinguish blue from black. The panels themselves are OK.

Response: Done.

L.148: suggest to rephrase “... technical design, resulting especially from the 3D ...”

Response: Revised accordingly.

*L.162: “converge *on* average values”*

Response: Fixed.

Figure 5: figure quality is low. And please increase line thickness in the legend.

Response: We replaced it with a vectorized version to improve the figure quality. Additionally, we

offered the original file in EPS format so that everything in the final paper (except for Extended Data Figure 8 which is too large to be saved in EPS, and we tried our best to make the JPG image clear) is vectorized and looks good when zoomed in. Like in Figure 2, we adjusted the legend to make it clearer.

*L.236: you probably wanted to say “results, increasing the risk of forecasting extreme weather events *inaccurately*” or “results, increasing the risk of underestimating the magnitudes of extreme weather events.”*

Response: Yes, and we changed it to “increasing the risk of underestimating the magnitudes of extreme weather events”. Thanks for making it clearer.

L.242: It might be good to add the references you provided in the rebuttal to the mentioning of 4D models.

Response: Agree. We moved the references to 4D deep networks to the main article and cited them accordingly in this part.

*L.247: “bring about *even* stronger performance.”*

Response: Done.

L.329 ff: Please add short definitions of the CRPS and the other new scores you have introduced.

Response: Done. We added a new paragraph titled “ensemble forecast metrics” to the appendix, after the paragraph of “evaluation metric”, where the definitions of the continuous ranked probability score (CRPS) and spread-skill ratio (SSR) were elaborated.

L.338: I suggest to rephrase “The RMSE and ACC metrics can also be evaluated for specific regions...”

Response: Done.

L.398: The special treatment could be described a bit more succinctly. For example: “... by half window size. Since coordinates in longitude direction are periodic, the half windows at the left and right edges are merged into one full window.”

Response: Revised accordingly. Thanks for the suggestion. The next sentence was also rephrased into: “The merge operation was not performed along the latitude direction because it is not periodic.”

L.435: two “limits”. Suggest: “the limited available computational budget prevented us from exploring this method.”

Response: Revised accordingly. Thanks!

L.467: “only look” (delete “s”)

Response: Fixed.

L.535: Parametrizations were not introduced primarily to “accelerate” but rather to achieve “closure”, i.e. solving the differential equations on scales that are smaller than the model resolution. Side note: there are interesting implications with respect to parametrizations when model resolution is increased substantially. For example, parametrizations of convection usually assume that all

transport of air occurs within one vertical column of grid boxes. However, once you go below ~5 km, you must take into account horizontal exchange between neighbouring boxes - and this means that convection directly impacts on the (spatially resolved) dynamics of the model. This is only one of the challenges for further advancing traditional NWP models.

Response: Agree. We revised the original sentence into: “The spacing of grids is key to forecast accuracy, but it is constrained by the computational budget and thus the spatial resolution of weather forecasts is often limited. Parameterization is an effective method for capturing unresolved processes.”

L.541: *suggestion: “offer a complementary path”.*

Response: Agree and fixed.

L.545: *“such as short-term precipitation...” - the point here is that on time scales up to a few hours, the traditional models are too much influenced by the initial conditions. Classical “nowcasting” methods have used traditional imaging algorithms (flow field extrapolation etc.), and these could be readily replaced by DL models.*

Response: Agree. We revised the sentence into: “AI-based methods were first applied to the problems of precipitation forecasting based on radar data or satellite data, where the traditional methods that are much influenced by the initial conditions were replaced by deep learning based methods.”

L.549: *I am now happy with the shorter discussion about NWP versus DL. However, I would suggest to add another sentence at the end referring to the current state of DL with very large networks achieving an abstract representation of the training data, and that this goes far beyond earlier attempts based on U-Nets or GANs, etc.*

Response: Agree. We appended a sentence to the end of the “Previous Work” section, saying: “State-of-the-art deep learning methods mostly rely on large models (i.e., with large numbers of learnable parameters) to learn complex patterns from the training data.”

L.636: *replace “studied in the experiments” with “used as inputs”.*

Response: Done.

L.658: *(Please include a definition of “tropics” in the figure caption. Did you use 15 or 20 degrees as latitude cutoff? Out of curiosity: some panels, especially T2m show an interesting zigzag pattern. Do you have an explanation for this?*

Response: We followed the definition of ECMWF and used 20 degrees as latitude cutoff. We added the definition of the tropics to the caption of Extended Data Figure 3: “We followed ECMWF to define the ‘tropics’ to be the region between latitude of +20° (inclusive) and –20° (inclusive).” We also emphasized in Extended Data Figures 1 and 2 that the Northern and Southern Hemispheres do not contain the tropics. The zigzag patterns have periods of 24 hours, implying that it is statistically easier to forecast weather variables of the same time-in-day because some weather factors (e.g., temperature) are largely impacted by the time in day.

Q3: *Extended data figures 4 and 5: to ease comparison, please add the 2018 results on these plots as*

well. (perhaps as dashed lines)

A3: Done.

Q4: *And finally a question: you mentioned that you ran Panguweather based on IFS initial states and would seek the reviewers' opinion on whether to include these results. However, I did not find them in the rebuttal letter nor the manuscript. In any case, I would suggest to leave the paper at its present state and only, if you so wish, include a sentence somewhere to say "Preliminary results with using IFS initial conditions instead of ERA5 data as model inputs suggest that Panguweather will lose some forecasting capability, but not deteriorate dramatically."*

A4: Thanks for the suggestion. We agree that it is better to leave the paper as its present state.

Response to Referee #2

Q1: I thank the authors for their replies and changes, I think the manuscript is significantly improved. Below I outline a few minor changes I think will improve the paper. After adopting these changes I would be happy to see the manuscript published.

A1: Thanks for helping us to improve the manuscript. We provide a point-by-point response below for the mentioned minor changes.

- *Figure 5: Please could the spread-skill plot in include a dotted/dashed horizontal line indicating a ratio of 1 to highlight the ideal behaviour, and mention this in the caption.*

Response: Done. We added an explanation in the caption, saying: “an ideal ensemble model produces spread-skill ratios of 1.0, displayed as the dashed lines”.

- *Extended figures 4 & 5 are very hard to interpret. I would suggest having one figure, showing years 2018, 2020 & 2021 in different colours to allow comparison. It should be noted on the plot that predictability varies from year to year which is not compensated for in this plot. If you have IFS data for all of these years then instead a better plot would be to show the performance relative to the IFS for each of the years, by comparing to the IFS baseline this would compensate for the predictability variation. e.g. plotting $(RMSE_{pangu} - RMSE_{ifs})/RMSE_{ifs}$ for each of the years separately. Even this has some caveats, as IFS has improved over this period, it would at least give context to the Pangu scores.*

Response: Done. We plotted the 2018, 2020 & 2021 data into the same figure. We do not have the IFS data in 2020 and 2021 from the ECMWF server, so we only reported the comparison using the 2018 data in Figure 2. We added a note in the caption of the merged figure saying: “It shall be noted that predictability varies from year to year which is not compensated for in this plot.”

- *Extended figure 8 appears to be very low resolution. I struggled to read it. I very much welcome it's inclusion though.*

Response: We replaced all figures with vectorized versions. Additionally, we submitted all figure files in EPS format (except for Extended Data Figure 8 which is too large to be saved in EPS, and we tried our best to make the JPG image clear) so that everything in the final paper can be vectorized.

- *Extended Figure 10 has no label.*

Response: We merged Extended Data Figures 10 and 11 (now it is Extended Data Figure 5) and added descriptions for each labeled panel.

- *L88: is this 16 days and 192 GPUs per model?*

Response: Each of the four models required 16 days on 192 GPUs. We made it clear in the paper. The revised sentence says: “Each of the four deep networks was trained for 100 epochs, and each of them takes approximately 16 days on a cluster of 192 NVIDIA Tesla-V100 GPUs.”

Q2: I cannot find several interesting details on your training. Specifically the loss used, the normalisation of the training data, and any scaling of the variables or pressure levels within the loss (or are all normalised variables given equal weighting?). I see no discussion of the learning rate used, or any details

on possible adaptive learning rate. Please specify this. Please include these.

A2: The details are explained here. We used the mean absolute error (MAE) loss. The normalization was performed on each 2D input field (e.g., Z500) separately. It worked by subtracting the mean value from the 2D field followed by dividing it by the standard deviation. The mean and standard deviation of each variable were computed on the weather data from 1979 to 2017. The weight for each variable was inversely proportional to the average loss value computed in an early run, which was designed to facilitate equivalence of the contributions by these variables. Specifically, the weights for upper-air variables were 3.00, 0.60, 1.50, 0.77, 0.54 for Z, Q, T, U, V, respectively, and the weights for surface variables were 1.50, 0.77, 0.66, 3.00 for MSLP, U10, V10, T2M, respectively. We added a weight of 1.0 to the MAE loss of the upper-air variables and 0.25 to that of the surface variables, and summed up the two losses. The learning rate started with 0.0005 and gradually annealed to 0 following the cosine schedule. All the above contents were updated to the “optimization details” paragraph.

Q3: *L438 DropPath*, please could the authors provide more detail about the specific implementation and settings that they used.

A3: We used the ScheduledDropPath^[a] with a drop ratio of 0.2. This is the standard implementation of the Swin transformer and we included it in our pseudocode. We added the elaboration and reference to the manuscript.

[a] Zoph, B., Vasudevan, V., Shlens, J., & Le, Q. V. Learning transferable architectures for scalable image recognition. In *Proc. IEEE/CVF Conf. Comput. Vis. Pattern Recognit.* 8697-8710 (2017).

Response to Referee #3

Q1: I would like to cordially thank the authors for responding in this new revision to my recommendation to make the models publicly available to the community - in addition to providing the original pseudo code which is also very helpful for a deeper understanding of the model.

Our group picked one trained model, the 24h forecast model, and I can confirm that it is very easy to download and run it. It just took us one afternoon to get this to work, and it executed quickly on even a desktop computer.

This means that anyone in the meteorological community can now run and test these models to their heart's desire. What a great opportunity for the community to explore how well the model predicts specific meteorological phenomena. Now THAT's going to help with progress in the field.

Kudos to the author for making this possible. Thank you!

A1: We thank the reviewer for the positive comments and all the help that helped us with improving the quality of the paper.

Reviewer Reports on the Second Revision:

Referees' comments:

Referee #2 (Remarks to the Author):

I thank the authors for their feedback and changes. I am happy with the updated manuscript and have only a couple of minor questions/comments.

- L29 approximate functions -> approximations. Or replace with "closure schemes"
- Could you state your batch size (apologies if this is already written and I have missed it).

Author Rebuttals to Second Revision:

Response to Referee #2

Q1: L29 approximate functions -> approximations. Or replace with "closure schemes"

A1: Thanks. We changed it to “approximations” accordingly.

Q2: Could you state your batch size (apologies if this is already written and I have missed it).

A2: Yes. We added a sentence to the “Optimization details” section in the appendix, saying: “We used a batch size of 192 (i.e., 1 training sample per GPU).”